# AuroraCap: Efficient, Performant Video Detailed Captioning and a New Benchmark

**Wenhao Chai** [1,2,*]   **Enxin Song** [*]   **Yilun Du** [4]   **Chenlin Meng** [2,3]   **Vashisht Madhavan** [2]
**Omer Bar-Tal** [2]   **Jenq-Neng Hwang** [1]   **Saining Xie** [5]   **Christopher D. Manning** [3]

[1] University of Washington   [2] Pika Labs   [3] Stanford   [4] Harvard   [5] New York University

Link: Leaderboard Video Detailed Caption | Model AuroraCap

## Abstract

Video detailed captioning is a key task which aims to generate comprehensive and coherent textual descriptions of video content, benefiting both video understanding and generation. In this paper, we propose AuroraCap, a video captioner based on a large multimodal model. We follow the simplest architecture design without additional parameters for temporal modeling. To address the overhead caused by lengthy video sequences, we implement the token merging strategy, reducing the number of input visual tokens. Surprisingly, we found that this strategy results in little performance loss. AuroraCap shows superior performance on various video and image captioning benchmarks, for example, obtaining a CIDEr of 88.9 on Flickr30k, beating GPT-4V (55.3) and Gemini-1.5 Pro (82.2). However, existing video caption benchmarks only include simple descriptions, consisting of a few dozen words, which limits research in this field. Therefore, we develop VDC, a video detailed captioning benchmark with over one thousand carefully annotated structured captions. In addition, we propose a new LLM-assisted metric VDC-score for bettering evaluation, which adopts a divide-and-conquer strategy to transform long caption evaluation into multiple short question-answer pairs. With the help of human Elo ranking, our experiments show that this benchmark better correlates with human judgments of video detailed captioning quality.

## 1 Introduction

The task of video detailed captioning involves generating comprehensive and coherent textual descriptions of video content, capturing not only the primary actions and objects but also intricate details, contextual nuances, and temporal dynamics. It has emerged as a critical area of research in computer vision and natural language processing, with significant implications for the fields of robotics (Yang et al., 2023b; Du et al., 2024), ego-centric perception (Grauman et al., 2022; 2023), embodied agents (Zhao et al., 2023; 2024c;d; Deng et al., 2023; 2024; Zhao et al., 2024b), and video editing (Chai et al., 2023) and generation (Bar-Tal et al., 2024). The challenges of video detailed captioning compared to past problems include the limited detailed caption data for training and evaluation, and also the lack of a good evaluation metric.

Before the emergence of Large Language Models (LLMs), previous models could only generate very short and rough descriptions of videos (Wang et al., 2022; Xu et al., 2023; Yan et al., 2022; Yang et al., 2023a). Although these models have been trained on web-scale video-text datasets (*e.g.,* HowTo100M (Miech et al., 2019) and VideoCC3M (Nagrani et al., 2022)), their capabilities remain limited due to their smaller scale and the lack of world knowledge possessed by LLMs. Recently, researchers start to build more powerful large multimodal models (LMMs) upon pretrained LLMs (*e.g.,* LLaVA (Liu et al., 2024b), InstructBlip (Dai et al., 2024b), InternVL (Chen et al., 2023b)). These models typically use intermediate components (*e.g.,* Q-Former (Li et al., 2023a) or an MLP) to connect the pre-trained vision transformer (ViT) (Dosovitskiy et al., 2020) and the LLM. Expanding from image-based LMMs to video-based LMMs is a natural progression, as videos can be viewed as sequences of frames. While most LMMs start with loading the pre-trained weights from image models and are further fine-tuned with video-text data, we find that LLaVA-like models can be easily adapted to a video one without any additional parameters but only with high-quality video-text instruction data for fine-tuning.

---

[*]Equal contributions.

Table 1: **Benchmark comparison** for video captioning task. Ave. Length indicates the average number of words per caption. Compared to the existing benchmarks, VDC has the longest captions.

| Dataset | Theme | # Video | # Clip | # Caption | # Word | # Vocab. | Ave. Length |
|---|---|---|---|---|---|---|---|
| MSVD (Chen & Dolan, 2011) | Open | 1,970 | 1,970 | 70,028 | 607,339 | 13,010 | 8.67 |
| MSR-VTT (Xu et al., 2016) | | 7,180 | 10,000 | 200,000 | 1,856,523 | 29,316 | 9.28 |
| ActivityNet (Krishna et al., 2017a) | | 20,000 | 100,000 | 100,000 | 1,340,000 | 15,564 | 13.40 |
| S-MiT (Monfort et al., 2021) | | 515,912 | 515,912 | 515,912 | 5,618,064 | 50,570 | 10.89 |
| M-VAD (Torabi et al., 2015) | Movie | 92 | 48,986 | 55,905 | 519,933 | 18,269 | 9.30 |
| MPII-MD (Rohrbach et al., 2013) | | 94 | 68,337 | 68,375 | 653,467 | 24,549 | 9.56 |
| Youcook2 (Zhou et al., 2018) | Cooking | 2,000 | 15,400 | 15,400 | 121,418 | 2,583 | 7.88 |
| Charades (Sigurdsson et al., 2016) | Human | 9,848 | 10,000 | 27,380 | 607,339 | 13,000 | 22.18 |
| VATEX (Wang et al., 2019) | | 41,300 | 41,300 | 413,000 | 4994,768 | 44,103 | 12.09 |
| VDC (ours) | Open | 1,027 | 1,027 | 1,027 | 515,441 | 20,419 | 500.91 |

However, naive treatment of videos as a series of image frames can result in significant computational overhead and may cause a generalization of the length problem (Wang et al., 2024b). To address these concerns, more specifically, to reduce the number of visual tokens, Video-LLaMA (Li et al., 2023f) adapts the video Q-former, MovieChat (Song et al., 2023) uses a memory bank, LLaMA-VID (Li et al., 2023g) simply uses global pooling, and FastV (Chen et al., 2024b) drops visual tokens by attention rank within LLM layers. In this paper, we present AURORACAP, adapting a simple yet efficient method called Token Merging (Bolya et al., 2022), which is proved to be effective in image and video classification and editing tasks (Li et al., 2023e). To be specific, we gradually combine similar tokens in a transformer layer using a bipartite soft matching algorithm to reduce the number of visual tokens. Following this pattern, our experiments show that we can use only 10% to 20% visual tokens compared to the original tokens generated by ViT with a marginal performance drop in various benchmarks. With this technique, it is easier to support higher-resolution and longer video sequence inputs for training and inference.

We present results on several widely used benchmarks, but find that existing video understanding benchmarks are either question-answer-based (Song et al., 2023; Chen & Dolan, 2011; Caba Heilbron et al., 2015; Xu et al., 2016; Xiao et al., 2021; Fu et al., 2024; Wu et al., 2024), which cannot demonstrate detailed descriptive abilities, or they provide descriptions that are too short, with only a few words (Xu et al., 2016; Caba Heilbron et al., 2015) as shown in Table 1. Some large-scale datasets focus on specific domains such as ego-centric (Grauman et al., 2023) or contain low-quality videos and annotations (Bain et al., 2021). Therefore, we construct the VDC (**V**ideo **D**etailed **C**aptions) benchmark, which contains over one thousand high-quality video-caption pairs spanning a wide range of categories, and the resulting captions encompass rich world knowledge, object attributes, camera movements, and crucially, detailed and precise temporal descriptions of events. We utilize `GPT-4o` as our recaption assistant with a hierarchical prompt design. To preserve as much information as possible from the videos and maintain temporal consistency, we implement a dense-frame extraction strategy. Using the dense frames as input, despite description of the whole video, we also generate high-quality captions from different aspects, including objective facts, backgrounds, camera angles and movements. Manual quality inspection is employed to ensure the quality of the video captions. While existing video-caption datasets (Chen et al., 2024d; Ju et al., 2024; Li et al., 2024b) offer structured captions, VDC is the first benchmark focused on detailed video captioning, providing significantly longer and more detailed captions than others as shown in the Table 1 and Section D.

We also introduce a novel evaluation metric specifically designed for detailed captioning task. Traditional metrics like METEOR (Banerjee & Lavie, 2005), CIDEr (Vedantam et al., 2015), and BLEU (Papineni et al., 2002), designed for machine translation or short captions, fail to evaluate detailed captions which contain rich semantic information. On the other side, an LLM-based evaluation metric is commonly used in visual question answering benchmarks (Maaz et al., 2023; Wu et al., 2024), especially for those generated by VLMs (Song et al., 2024; 2023). However, we observe that the LLM-based evaluation metric still struggles to differentiate the quality of detailed captions and tends to give lower scores. To address these challenges, we propose VDCSCORE, a novel captioning evaluation metric that leverages the reliability of large language models (LLMs) by evaluating short visual question-answer pairs. We first decompose the ground-truth caption into a set of concise question-answer pairs using the LLM, then generate corresponding responses from the

predicted caption. Finally, the LLM is used to assess the accuracy of each response to provide an overall score. In particular, our paper makes the following contributions:

- In Section 2, we illustrate how we can reduce the number of tokens used for image or video input before injecting into LLM with marginal performance drop. Using these insights, we propose AURORACAP, a video understanding base model with improved captioning performance, which also demonstrates advanced performance on existing benchmarks.
- In Section 3, we present VDC, the first benchmark for detailed video captioning, featuring over one thousand videos with significantly longer and more detailed captions. We comprehensively evaluate proprietary and open-source models using our proposed VDCSCORE metric. We also conduct extensive ablation studies based on AURORACAP.

## 2 AURORACAP: A VIDEO DETAILED CAPTIONING BASELINE

### 2.1 ARCHITECTURE

**Large multimodal model.** To effectively leverage the capabilities of both the pre-trained LLM and visual model, which is typically CLIP (Radford et al., 2021) or DINO (Oquab et al., 2023), LLaVA adapt a simple multi-layer perceptron (MLP) as projection layer to connect each patch tokens of image features into the word embedding space. The original LLaVA model is trained by a two-stage instruction-tuning procedure, which first pretraining projection layer for feature alignment and then finetuning end-to-end while freeze the visual encoder. Recent works like Prismatic VLMs (Karamcheti et al., 2024) and Idefics2 (Laurençon et al., 2024) further explore the design space of LLaVA architecture. We adapt some conclusion from those works for training the model.

**Token merging.** To increase the throughput of existing ViT models, Token Merging (Bolya et al., 2022) is proposed to gradually combines similar tokens in a transformer to reduce the number of tokens passing through ViT models. Token Merging has been proven to be effective on image and video classification tasks even without the need for training. Token Merging is applied between the attention and MLP within each transformer block as:

1. Alternatively partition the tokens into two sets $\mathbb{A}$ and $\mathbb{B}$ of roughly equal size.
2. For each token in set $\mathbb{A}$, calculate the token similarity with each token in set $\mathbb{B}$ based on cosine similarity of the *Key* features in attention block.
3. Use bipartite soft matching and then select the most similar $r$ pairs.
4. Merge the tokens using weighted average, record the token size.
5. Concatenate the two sets $\mathbb{A}$ and $\mathbb{B}$ back together again.

Once the tokens have been merged, they actually carry the features for more than one input patch. Therefore, the merged tokens will have less effect in softmax attention as

$$\mathbf{A} = \text{softmax}\left(\frac{\mathbf{QK}^\top}{\sqrt{\mathbf{d}}} + \log \mathbf{s}\right) \tag{1}$$

where $\mathbf{s}$ is the number of patches the token represents after token merging. We conduct frame-wise token merging in AURORACAP, the visualization of token merging can be found in Appendix E.

### 2.2 TRAINING RECIPE

Building upon the exploration in works like Prismatic VLMs (Karamcheti et al., 2024), Idefics2 (Laurençon et al., 2024), and InternVL (Chen et al., 2023b), we further adopt a three-stage training strategy, which can be noted as Pretraining stage, Vision stage and Language stage. The training data used in each stage are shown in Table G6, Table G7 and Table G8. More training details including hyper-parameters selection and data preprocessing operation can be found in Appendix G.

**Pretraining stage.** Similar to LLaVA, we first align visual features from the vision encoder ViT with the word embedding space of LLMs. To achieve this, we freeze the pretrained ViT and LLM, training solely the vision-language connector. Consistent with LLaVA-1.5 (Li et al., 2024a), we employ a two-layer MLP as the projection layer and pretrain on 1.3M image-caption pairs. To optimize performance, we explore various combinations of the pre-trained ViT and LLM in Appendix F.

Table 2: Comparison of AURORACAP with LLM-based SoTA methods on image-captioning benchmarks under a zero-shot setting, except $k$-shot where $k$ is marked as a superscript. The GPT-4V CIDEr results are from (Team, 2024), the only reference we know of, but seem oddly low. ∗ marks fine-tuned results. Scores in **bold** indicate the best performance under zero-shot setting.

| Model | Flickr (31,784) | | | | | NoCaps (4,500) | | | | | COCO-Cap (5,000) | | | | |
|---|---|---|---|---|---|---|---|---|---|---|---|---|---|---|---|
| | CIDEr | BLEU@1 | BLEU@4 | Meteor | ROUGE | CIDEr | BLEU@1 | BLEU@4 | Meteor | ROUGE | CIDEr | BLEU@1 | BLEU@4 | Meteor | ROUGE |
| InstructBLIP-7B (Dai et al., 2024b) | 78.2 | 77.1 | 30.9 | 24.8 | 53.4 | 118.2 | 88.2 | 47.2 | 30.3 | 62.0 | 141.3* | 82.8* | 41.7* | 30.9* | 61.0* |
| InstructBLIP-13B (Dai et al., 2024b) | 76.1 | 76.8 | 30.1 | 24.4 | 53.0 | 116.3 | 88.3 | 46.7 | 29.8 | 61.4 | 135.0* | 82.2* | 39.7* | 29.8* | 59.8* |
| LLaVA-1.5-7B (Liu et al., 2023) | 74.9 | 71.7 | 28.4 | 26.1 | 52.8 | 105.5 | 82.6 | 40.2 | 30.3 | 59.4 | 110.3 | 73.0 | 29.7 | 29.2 | 55.5 |
| LLaVA-1.5-13B (Liu et al., 2023) | 79.4 | 73.6 | 30.2 | 26.6 | 53.9 | 109.2 | 84.2 | 42.4 | 30.6 | 60.3 | 115.6 | 74.6 | 31.5 | 29.4 | 56.5 |
| LLaVA-1.6-7B (Liu et al., 2024a) | 68.4 | 69.6 | 26.6 | 23.2 | 50.3 | 88.4 | 73.8 | 34.8 | 25.9 | 54.6 | 99.9 | 67.7 | 28.4 | 25.5 | 52.4 |
| LLaVA-1.6-13B (Liu et al., 2024a) | 66.6 | 65.2 | 24.2 | 22.2 | 48.8 | 88.1 | 68.7 | 34.0 | 25.4 | 54.9 | 101.8 | 62.2 | 27.5 | 24.6 | 52.1 |
| MiniCPM-V-3B (Hu et al., 2023) | 66.8 | 68.0 | 25.1 | 27.2 | 51.0 | 89.9 | 79.1 | 33.2 | 29.7 | 55.8 | 94.2 | 69.8 | 23.9 | 28.3 | 52.3 |
| DeCap (Li et al., 2023d) | 56.7 | – | 21.2 | 21.8 | – | 42.7 | – | – | – | – | 91.2 | – | 24.7 | 25.0 | – |
| Flamingo-80B (Alayrac et al., 2022) | 67.2 | – | – | – | – | – | – | – | – | – | 84.3 | – | – | – | – |
| Chameleon-34B (Team, 2024) | 74.7² | – | – | – | – | – | – | – | – | – | 120.2² | – | – | – | – |
| GPT-4V | 55.3⁸ | – | – | – | – | – | – | – | – | – | 78.5⁸ | – | – | – | – |
| Gemini-1.5 Pro | 82.2⁴ | – | – | – | – | – | – | – | – | – | 99.8² | – | – | – | – |
| AURORACAP-7B | 88.9 | 75.6 | 32.8 | 26.7 | 55.4 | 111.4 | 85.6 | 44.4 | 29.9 | 60.6 | 120.8 | 78.0 | 35.3 | 28.6 | 57.2 |

Table 3: Comparison of AURORACAP with SoTA methods on existing video captioning benchmarks under zero-shot setting. AURORACAP substantially outperforms other recent methods.

| Model | MSR-VTT (1,000) | | | | | VATEX (1,000) | | | | |
|---|---|---|---|---|---|---|---|---|---|---|
| | CIDEr | BLEU@1 | BLEU@4 | Meteor | ROUGE | CIDEr | BLEU@1 | BLEU@4 | Meteor | ROUGE |
| ZeroCap (Tewel et al., 2022) | 9.6 | – | 2.9 | 16.3 | 35.4 | – | – | – | – | – |
| DeCap (Li et al., 2023d) | 18.6 | – | 14.7 | 20.4 | – | 18.7 | – | 13.1 | 15.3 | – |
| PaLI-3 (Chen et al., 2022) | 21.3 | – | – | – | – | – | – | – | – | – |
| Ma *et al.* (Ma et al., 2024) | 22.1 | – | 3.5 | 17.3 | 28.7 | 23.9 | – | 2.8 | 14.1 | 23.5 |
| LLaVA-7B (Liu et al., 2024b) | 16.9 | – | – | – | – | – | – | – | – | – |
| Video-LLaMA (Zhang et al., 2023a) | 2.3 | – | 4.9 | 16.8 | – | 3.8 | – | 4.3 | 16.3 | 21.8 |
| AURORACAP-7B | 33.1 | 58.6 | 21.0 | 23.9 | 49.5 | 33.8 | 57.1 | 18.4 | 19.0 | 40.8 |

**Vision stage.** Unlike LLaVA, we next unfreeze the pretrained ViT while freezing the LLM during vision stage and train with the public data among various computer vision tasks (*e.g.,* captioning, object identification, classification, reasoning, VQA, *etc.*) to get better generalization (Huh et al., 2024). The motivation for doing this is that CLIP ViT usually performs poorly in aspects such as Orientation and Direction, Positional and Relational Context, Quantity and Count (Tong et al., 2024b). However, since the most of the collected datasets lack high-quality and detailed corresponding language descriptions, the labels often consist of only a few words or a short phrase when converted to text. Therefore, unfreezing the language model at this stage is risky, as it may lead to a degradation in the performance of the language model.

**Language stage.** Finally, we conduct end-to-end training, which means all the components are trainable, with the most high-quality public data during language stage training. We mix all the data, including images and videos, captions and instructions, into each mini-batch for training. To improve video captioning performance, we duplicate the video captioning datasets twice. We remove all the video training data for training a image-based AURORACAP as well for image captioning task.

## 2.3 EVALUATION

In this section we evaluate AURORACAP on various tasks including image captioning, video captioning, and video question answering. Appendix H show detailed evaluation settings.

**Image Captioning.** We evaluate AURORACAP using CIDEr (C), BLEU-4 (B@4), BLEU-1 (B@1), METEOR (M), and ROUGE-L (R) metric on Flickr (Plummer et al., 2015), NoCaps (Agrawal et al., 2019), and COCO-Cap (Lin et al., 2014) benchmarks and compare it with LLM-based state-of-the-art methods as shown in Table 2. We show the performance of image based AURORACAP under zero-shot settings. Notice that these benchmarks all contain short captions consisting of a single sentence, so they only partially reflect the model's performance. The performance mentioned in the rest of this paper refers to video-based AURORACAP.

**Video Captioning.** Although the current video captioning benchmarks only contain one-sentence captions, to compare with prior work, we similarly evaluate on these benchmarks. We evaluate AURORACAP using CIDEr (C), BLEU-4 (B@4), BLEU-1 (B@1), METEOR (M), and ROUGE-L (R) metric on MSR-VTT (Xu et al., 2016), VATEX (Wang et al., 2019) as shown in Table 3.

Table 4: Comparison of AURORACAP with SoTA methods on video question answering and classification benchmarks under zero-shot setting. The pretrained LLM size is 7B for all listed models.

| Model | ANet | | MSVD | | MSR-VTT | | iVQA |
|---|---|---|---|---|---|---|---|
| | Acc | Score | Acc | Score | Acc | Score | Acc |
| Just Ask (Yang et al., 2021) | – | – | – | – | – | – | 12.2 |
| FrozenBiLM (Yang et al., 2022) | 24.7 | – | 32.2 | – | 16.8 | – | 26.8 |
| Video-LLaMA (Zhang et al., 2023a) | 12.4 | 1.1 | 51.6 | 2.5 | 29.6 | 1.8 | – |
| VideoChat (Li et al., 2023b) | 26.5 | 2.2 | 56.3 | 2.8 | 45.0 | 2.5 | – |
| Video-ChatGPT (Maaz et al., 2023) | 35.2 | 2.7 | 64.9 | 3.3 | 49.3 | 2.8 | – |
| LLaMA-VID (Li et al., 2023g) | 47.4 | 3.3 | 69.7 | 3.7 | 57.7 | 3.2 | – |
| Video-LLaVA (Lin et al., 2023a) | 45.3 | 3.3 | 70.7 | 3.9 | 59.2 | 3.5 | – |
| FreeVA (Wu, 2024) | 51.2 | 3.5 | 73.8 | **4.1** | **60.0** | **3.5** | – |
| LLaVA-NeXT-Video (Liu et al., 2024a) | 53.5 | 3.2 | – | – | – | – | – |
| MovieChat (Song et al., 2023) | 45.7 | 3.4 | 75.2 | 3.8 | 52.7 | 2.6 | – |
| MovieChat+ (Song et al., 2024) | 48.1 | 3.4 | **76.5** | 3.9 | 53.9 | 2.7 | – |
| AURORACAP-7B | **61.8** | **3.8** | 62.6 | 3.6 | 43.5 | 2.9 | **55.2** |

**Video Question Answering.** We evaluate AURORACAP on MSVD-QA (Xu et al., 2017), ActivityNet-QA (Yu et al., 2019), MSRVTT-QA (Xu et al., 2017), and iVQA (Yang et al., 2021) for video question answering tasks as shown in Table 4. Although AURORACAP is primarily a captioning model, it achieves competitive performance in some VQA datasets (ANet, iVQA). For others (MSVD, MSR-VTT) performance is more modest, but still not bad. In some failure cases observed in the model, we found that prompting the model to generate a comprehensive caption for the video input can lead to outputs that include the correct answer. This phenomenon may be attributed to a disruption in the model's instruction-following capabilities during the training.

## 3 VDC: A VIDEO DETAILED CAPTIONING BENCHMARK

### 3.1 BENCHMARK DATASET CURATION

#### 3.1.1 VIDEO COLLECTION AND PROCESSING

To ensure the reliability of the benchmark, it is crucial to maintain high video quality, balanced data distribution, and content complexity. Panda-70M (Chen et al., 2024e) offers a high-resolution, open-domain YouTube video dataset with diverse one-minute clips across wildlife, cooking, sports, news, TV shows, gaming, and 3D rendering, ideal for studying complex real-world scenarios. Additionally, a large volume of aesthetically appealing videos from user-uploaded platforms like Mixkit (Mixkit, 2024), Pixabay (Pixabay, 2024), and Pexels (Pexels, 2024) provides scenic views and visually pleasing human activities with minimal transitions and simpler events. Ego4D (Grauman et al., 2022) complements the video source by focusing on ego-centric human activities and auto-driving scenarios, ensuring comprehensive coverage of real-world scenes. To mitigate content homogeneity among these candidate videos and maintain diversity in the final dataset, inspired by ShareGPT4Video (Chen et al., 2024d), we building VDC upon those various video sources. Note that the videos used in VDC construction are not included in the training data of AURORACAP. To ensure balanced data distribution, we allocate equal proportions of videos from Panda-70M (Chen et al., 2024e), Ego4D (Grauman et al., 2022), Mixkit (Mixkit, 2024), Pixabay (Pixabay, 2024), and Pexels (Pexels, 2024). We first split the video into clips and apply dense frame extraction.

#### 3.1.2 STRUCTURED DETAILED CAPTIONS CONSTRUCTION PIPELINE

We believe that a comprehensive detailed video caption benchmark should encompass various aspects, including main objects, camera movements, and background. However, most existing benchmarks (Li et al., 2024b; Chen et al., 2024d) provide only a single caption for the entire video with less structured details. Therefore, we develop a structured detailed captions construction pipeline to generate extra detailed descriptions from various perspectives, significantly extending the length and enhancing the richness compared to previous benchmarks. Following (Ju et al., 2024), the structured captions

Table 5: The video source distribution of the proposed VDC benchamrk including diverse settings such as natural landscapes, human activities, and animal activities. Videos from different sources have a similar proportion in VDC, reducing the data bias.

| Video Source | # Sample | Proportion | Duration (sec.) | Ave. Length (sec.) | Ave. # Keyframe |
|---|---|---|---|---|---|
| Panda-70M (Chen et al., 2024e) | 229 | 22.25% | 5,714 | 24.95 | 7.18 |
| Ego4D (Grauman et al., 2022) | 196 | 19.05% | 10,935 | 55.79 | 19.46 |
| Mixkit (Mixkit, 2024) | 197 | 19.14% | 3,261 | 16.55 | 6.58 |
| Pixabay (Pixabay, 2024) | 199 | 19.34% | 4,748 | 23.86 | 8.99 |
| Pexels (Pexels, 2024) | 208 | 20.21% | 4,343 | 20.88 | 7.99 |
| Total | 1,027 | – | 29,001 | 28.18 | 10.43 |

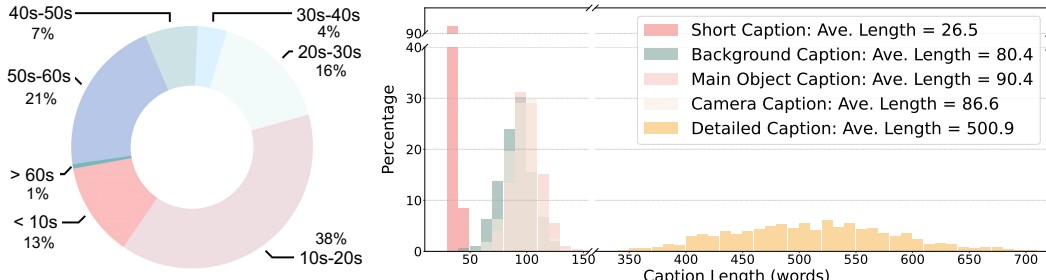

Figure 1: Video length statistics.          Figure 2: Distribution of structured caption length.

in VDC encompass not only **short** and **detailed captions** but also three additional categories: (1) **main object caption**, offering a comprehensive analysis of the primary subjects' actions, attributes, interactions, and movements across frames, including variations in posture, expression, and speed; (2) **background caption**, providing detailed descriptions of the background, such as objects, location, weather, time, and dynamic elements; and (3) **camera caption**, which details the camera work, including shot types, angles, movements, transitions, and special effects.

To generate detailed, fine-grained, and accurate captions, we leverage `GPT-4o` to produce video descriptions. We utilize the dense video frames to obtain captions. We observed that generating all captions in a single conversation round often introduces hallucinations in the detailed captions. To address this, we design a hierarchical prompt strategy to efficiently obtain accurate structured captions and detailed captions in two conversation rounds: (1) structured captions generation and (2) detailed captions integration. In the first round, the prompt briefly introduces the differences between structured captions and uses the dense video frames as input to generate the short caption, main object caption, background caption, camera caption, and the detailed caption. In the second round, the generated captions serve as the reference. The second-round prompt guides `GPT-4o` to enhance the detailed caption based on the initial captions, ensuring consistency without introducing new entities or relations, and producing a vivid, engaging, and informative description. The whole prompt template for the structured detailed captions construction pipeline can be found in Appendix K. Finally, we conduct a manual review to correct captions with hallucinations and supplement omitted visual elements. The refined detail structured captions are then used as the ground truth for evaluation.

### 3.1.3 COMPARISON ON NUMERICAL STATISTICS

Based on the hierarchical scheme, VDC can capture a rich variety of details of the video and reduce hallucinations. The visual representation in Figure 1 demonstrates the video duration distribution of VDC. Over $87\%$ of the videos exhibit a duration ranging from 10K to 12K frames, while $1\%$ of videos extending beyond 60 seconds. Only $13\%$ of videos have duration less than 10 seconds. As illustrated in Table 1, the average length of detailed descriptions in VDC is significantly longer than in previous benchmarks. Figure 2 shows the length distribution of structured captions in VDC, with detailed captions averaging over 500 words. Appendix M present more statistics.

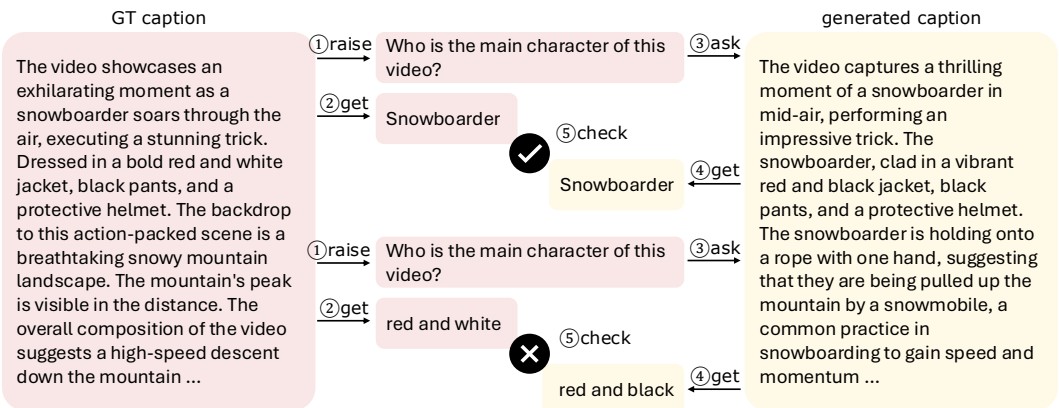

Figure 3: Evaluation pipeline with VDCSCORE. Like when humans take reading comprehension tests, we transform the matching between two paragraphs into a set of question-answer pairings. We first generate some question-answer pairs based on the ground truth captions, then derive corresponding answers one by one from the generated captions, and finally perform matching. The process is automatically evaluated with the LLM involvement in each step.

## 3.2 EVALUATION METRIC DESIGN AND LEADERBOARD

### 3.2.1 VDCSCORE: EVALUATING DETAILED CAPTIONS WITH LLMS

Evaluating video captions requires not only assessing the quality of the captions but also flexibly evaluating the alignment between the video and the caption. While metrics such as BLEU (Papineni et al., 2002), CIDEr (Vedantam et al., 2015), and ROUGE-L (Lin, 2004) have been employed for caption evaluation, these metrics are predominantly designed for short captions and rely heavily on word-level frequency-based alignment. Given the advanced semantic understanding capabilities of large language models (LLMs), Video-ChatGPT (Li et al., 2023b) proposes using LLM as an evaluation assistant to directly judge the correctness of the whole predicted captions and assign scores. However, as demonstrated in Table 6, our experiments indicate that when dealing with detailed captions, the direct application of LLM struggles to accurately distinguish the correctness of various predicted captions, fails to effectively evaluate the precision of detailed descriptions, and exhibits a tendency to assign disproportionately lower scores. Therefore, we introduce VDCSCORE, a novel quantitative metric that utilizes LLMs to evaluate the similarity between predicted and ground-truth detailed captions through a divide-and-conquer approach.

The core idea of VDCSCORE is to decompose long detailed captions into multiple short question-answering pairs, avergae the evaluation of each pair as the final result. We elaborate the design of VDCSCORE in the following parts: (1) ground-truth question-answer pairs extraction, (2) responsed answers generation and (3) answers matching. As illustrated in Figure 3, we first employ GPT-4o* to generate question-answer pairs from the detailed ground-truth caption. To ensure that the generated question-answer pairs capture as much information as possible from the original caption and facilitate accurate evaluation by LLMs, we constrain the number of generated pairs and impose specific guidelines: the questions must be open-ended, and the answers should be concise, and directly relevant to the questions. For a fair comparison and to mitigate potential variability arising from generating different question-answer pairs for the same caption, we pre-generate a standardized set of question-answer pairs for all captions in VDC, as depicted in Figure 3. The used prompt templates used along with additional examples, are provided in Appendix L.

VDCSCORE subsequently analyzes the predicted captions by leveraging ground-truth question-answer pairs. We prompt GPT-4o to read the detailed predicted captions and generate answers based solely on these captions. To mitigate biases arising from discrepancies in the length between ground-truth and predicted answers, we also impose constraints ensuring that responses are limited to concise sentences or phrases. Consequently, for each pair of ground-truth and predicted captions, we

---

*gpt-4o-2024-08-06, the latest GPT-4o version in September, 2024.

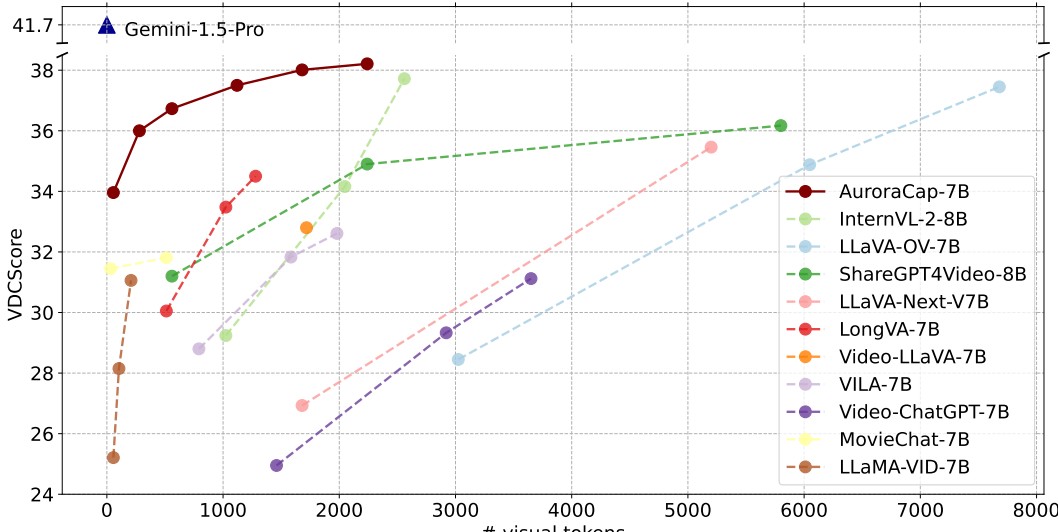

Figure 4: Comparison between various models with different number of visual tokens input on VDC. For Gemini-1.5-Pro, we only report the performance. We manage the number of visual tokens by managing token merging for AURORACAP, and manage the number of frames for others. AURORA-CAP achieves a much better VDCSCORE than all other models given a certain compression in the number of visual tokens kept and indeed approaches the performance of Gemini-1.5-Pro.

Table 6: Comparison of AURORACAP with LLM-based baseline methods on VDCSCORE and other evaluation metrics under zero-shot setting. For each evaluation metric, we report the average value of the five structured captions in VDC. Note that VDD, CIDEr, and BLEU are only the average of background and main object caption, since the values of the others are closed to zero.

| Model | VDCSCORE | | VDD | | CI-DEr | BLEU @1 | BLEU @4 | Met-eor | ROU-GE | Elo |
|---|---|---|---|---|---|---|---|---|---|---|
| | Acc | Score | Acc | Score | | | | | | |
| Gemini-1.5 Pro (Reid et al., 2024) | 41.73 | 2.15 | 49.68 | 3.07 | 5.97 | 29.72 | 2.63 | 21.21 | 20.19 | 1,171 |
| LLaMA-VID (Li et al., 2023f) | 30.86 | 1.62 | 4.63 | 1.63 | 1.48 | 17.74 | 1.46 | 8.07 | 17.47 | 859 |
| Video-ChatGPT-7B (Maaz et al., 2023) | 31.12 | 1.62 | 8.57 | 1.84 | 2.92 | 17.31 | 2.19 | 11.57 | 16.96 | 944 |
| MovieChat-7B (Song et al., 2023) | 31.92 | 1.64 | 10.24 | 1.86 | 5.14 | 14.33 | 3.17 | 13.60 | 14.98 | 890 |
| VILA-7B (Lin et al., 2023b) | 32.61 | 1.70 | 16.27 | 2.02 | 8.20 | 19.13 | 2.11 | 5.62 | 16.63 | 1,073 |
| Video-LLaVA-7B (Lin et al., 2023a) | 32.80 | 1.72 | 14.14 | 2.00 | 4.43 | 17.20 | 2.32 | 10.36 | 17.53 | 1,007 |
| LLaVA-1.5-7B (Liu et al., 2023) | 33.98 | 1.76 | 26.71 | 2.33 | 6.63 | 29.80 | 2.54 | 22.79 | 20.36 | 825 |
| LongVA-7B (Zhang et al., 2024b) | 34.50 | 1.79 | 32.65 | 2.69 | 4.83 | 18.75 | 2.16 | 13.43 | 14.84 | 969 |
| LLaVA-1.5-13B (Liu et al., 2023) | 34.78 | 1.80 | 28.26 | 2.36 | 3.90 | 20.43 | 2.02 | 26.37 | 17.87 | 943 |
| LLaVA-NeXT-V7B (Zhang et al., 2024d) | 35.46 | 1.85 | 25.62 | 2.34 | 2.66 | 20.18 | 2.33 | **28.17** | 17.51 | 1,022 |
| LLaVA-1.6-7B (Liu et al., 2024a) | 35.70 | 1.85 | 40.16 | 2.69 | 3.09 | 17.36 | 1.59 | 24.23 | 17.08 | 846 |
| LLaVA-1.6-13B (Liu et al., 2024a) | 35.85 | 1.85 | 34.55 | 2.51 | 5.55 | 29.23 | 2.50 | 20.26 | 19.96 | 728 |
| ShareGPT4Video-8B (Chen et al., 2024d) | 36.17 | 1.85 | 36.44 | 1.85 | 1.02 | 12.61 | 0.79 | 8.33 | 16.31 | 1,102 |
| LLaVA-OV-7B (Li et al., 2024a) | 37.45 | 1.94 | 41.83 | 2.70 | 4.09 | 28.34 | 2.84 | 23.98 | 19.59 | 1,155 |
| InternVL-2-8B (Chen et al., 2023b) | 37.72 | 1.96 | **48.99** | **3.03** | 5.59 | 15.75 | 2.48 | 10.76 | 17.63 | 1,081 |
| AURORACAP-7B | **38.21** | **1.98** | 48.33 | 2.90 | **9.51** | **30.90** | **4.06** | 19.09 | **21.58** | **1,267** |

obtain a set of `<question, correct answer, predicted answer>` triplets. Following Video-ChatGPT (Li et al., 2023b), we then ask `GPT-4o` to output two scores for each triplet: one for answer correctness and another for answer quality. The final accuracy and score are calculated by averaging the correctness score and quality score respectively. When using two same captions as input, VDCSCORE returns an accuracy of 100%, demonstrating the feasibility and reliability.

### 3.2.2 BENCHMARKING VIDEO DETAILED CAPTIONING

To our knowledge, no standard evaluation benchmarks have been established for detailed video captioning. To advance this field, we assess several baselines on our proposed VDC. As illustrated in Table 6, we present a quantitative comparison between our AURORACAP with existing state-of-the-art LMMs. We compare the VDCSCORE with both rule-based and model-based caption metrics with AURORACAP performing well. BLEU (Papineni et al., 2002), CIDEr (Vedantam et al.,

Table 7: Comparison of AURORACAP with LLM-based baseline methods on VDCSCORE under zero-shot structured captions setting. We consider the VDCSCORE of detailed captions, short captions, background captions, main object captions and camera captions. We also test the vision-blind case suggesting by (Chen et al., 2024c; Tong et al., 2024a).

| Model | Camera Acc / Score | Short Acc / Score | Background Acc / Score | Main Object Acc / Score | Detailed Acc / Score |
|---|---|---|---|---|---|
| Vicuna-v1.5-7B (Chiang et al., 2023) | 21.68 / 1.12 | 23.06 / 1.17 | 22.02 / 1.15 | 22.64 / 1.16 | 23.09 / 1.20 |
| Llama-3.1-8B (Dubey et al., 2024) | 17.83 / 1.00 | 17.90 / 1.02 | 19.52 / 1.10 | 19.57 / 1.10 | 20.10 / 1.22 |
| Gemini-1.5 Pro (Reid et al., 2024) | 38.68 / 2.05 | 35.71 / 1.85 | 43.84 / 2.23 | 47.32 / 2.41 | 43.11 / 2.22 |
| LLaMA-VID (Li et al., 2023f) | 39.47 / 2.10 | 29.92 / 1.56 | 28.01 / 1.45 | 31.24 / 1.59 | 25.67 / 1.38 |
| Video-ChatGPT-7B (Maaz et al., 2023) | 37.46 / 2.00 | 29.36 / 1.56 | 33.68 / 1.70 | 30.47 / 1.60 | 24.61 / 1.26 |
| MovieChat-7B (Song et al., 2023) | 37.25 / 1.98 | 32.55 / 1.59 | 28.99 / 1.54 | 31.97 / 1.64 | 28.82 / 1.46 |
| VILA-7B (Lin et al., 2023b) | 34.33 / 1.83 | 30.40 / 1.55 | 35.15 / 1.80 | 33.38 / 1.72 | 29.78 / 1.58 |
| Video-LLaVA-7B (Lin et al., 2023a) | 37.48 / 1.97 | 30.67 / 1.63 | 32.50 / 1.70 | 36.01 / 1.85 | 27.36 / 1.43 |
| LLaVA-1.5-7B (Liu et al., 2023) | 38.38 / 2.04 | 28.61 / 1.51 | 34.86 / 1.79 | 34.62 / 1.76 | 33.43 / 1.73 |
| LongVA-7B (Zhang et al., 2024b) | 35.32 / 1.90 | 31.94 / 1.63 | 36.39 / 1.85 | 40.95 / 2.11 | 27.91 / 1.48 |
| LLaVA-1.5-13B (Liu et al., 2023) | 38.97 / 2.07 | 30.89 / 1.60 | 34.79 / 1.78 | 36.27 / 1.84 | 33.00 / 1.74 |
| LLaVA-NeXT-V7B (Zhang et al., 2024d) | 39.73 / 2.10 | 30.63 / 1.60 | 36.54 / 1.88 | 36.54 / 1.88 | 33.84 / 1.77 |
| LLaVA-1.6-7B (Liu et al., 2024a) | 36.50 / 1.93 | 31.91 / 1.65 | 37.58 / 1.92 | 36.03 / 1.85 | 36.47 / 1.89 |
| LLaVA-1.6-13B (Liu et al., 2024a) | 35.61 / 1.86 | 31.90 / 1.66 | **38.90 / 1.99** | 36.65 / 1.87 | 36.18 / 1.89 |
| ShareGPT4Video-8B (Chen et al., 2024d) | 33.28 / 1.76 | **39.08 / 1.94** | 35.77 / 1.81 | 37.12 / 1.89 | 35.62 / 1.84 |
| LLaVA-OV-7B (Li et al., 2024a) | 37.82 / 2.02 | 32.58 / 1.70 | 37.43 / 1.92 | 38.21 / 1.96 | 41.20 / 2.13 |
| InternVL-2-8B (Chen et al., 2023b) | 39.08 / 2.11 | 33.02 / 1.74 | 37.47 / 1.89 | **44.16 / 2.22** | 34.89 / 1.82 |
| AURORACAP-7B | **43.50 / 2.27** | 32.07 / 1.68 | 35.92 / 1.84 | 39.02 / 1.97 | **41.30 / 2.15** |

Table 8: Comparison of AURORACAP with LLM-based baseline methods on VDD (Liu et al., 2024a) under zero-shot structured captions setting. We consider the VDD (Liu et al., 2024a) of detailed captions, short captions, background captions, main object captions and camera captions.

| Model | Camera Acc / Score | Short Acc / Score | Background Acc / Score | Main Object Acc / Score | Detailed Acc / Score |
|---|---|---|---|---|---|
| Gemini-1.5 Pro (Reid et al., 2024) | 18.89 / 2.115 | 16.91 / 1.572 | 57.73 / 3.263 | 41.64 / 2.886 | 2.581 / 0.330 |
| LLaMA-VID (Li et al., 2023f) | 28.28 / 2.513 | 1.034 / 1.042 | 6.198 / 1.895 | 3.063 / 1.366 | 2.046 / 0.304 |
| Video-ChatGPT-7B (Maaz et al., 2023) | 16.00 / 2.175 | 4.173 / 1.032 | 14.14 / 2.273 | 3.001 / 1.423 | 1.038 / 0.192 |
| VILA-7B (Lin et al., 2023b) | 4.005 / 1.751 | 2.087 / 0.233 | 22.45 / 2.385 | 10.10 / 1.672 | 1.015 / 0.262 |
| Video-LLaVA-7B (Lin et al., 2023a) | 20.00 / 2.336 | 3.193 / 1.064 | 17.17 / 2.253 | 11.11 / 1.765 | 3.130 / 0.316 |
| LLaVA-1.5-7B (Liu et al., 2023) | 26.88 / 2.515 | 1.222 / 0.793 | 36.50 / 2.725 | 16.92 / 1.937 | 0.694 / 0.276 |
| LongVA-7B (Zhang et al., 2024b) | 17.00 / 2.204 | 1.016 / 0.794 | 50.00 / 3.203 | 15.31 / 2.196 | 0.002 / 0.247 |
| LLaVA-1.5-13B (Liu et al., 2023) | 32.65 / 2.662 | 2.836 / 0.922 | 37.55 / 2.749 | 18.98 / 1.978 | 0.688 / 0.275 |
| LLaVA-NeXT-V7B (Zhang et al., 2024d) | 29.81 / 2.645 | 1.913 / 0.957 | 33.23 / 2.692 | 18.02 / 1.999 | 0.887 / 0.279 |
| LLaVA-1.6-7B (Liu et al., 2024a) | 21.11 / 2.229 | 8.696 / 1.146 | 53.31 / 3.105 | 27.01 / 2.286 | 1.282 / 0.279 |
| LLaVA-1.6-13B (Liu et al., 2024a) | 21.56 / 2.199 | 9.798 / 1.206 | 39.37 / 2.692 | 29.73 / 2.329 | 1.287 / 0.271 |
| ShareGPT4Video-8B (Chen et al., 2024d) | 33.28 / 1.768 | 4.908 / 0.986 | 35.77 / 1.813 | 37.12 / 1.899 | **3.213 / 0.752** |
| LLaVA-OV-7B (Li et al., 2024a) | 17.11 / 2.086 | **11.15 / 1.277** | 55.82 / 3.149 | 27.84 / 2.258 | 1.372 / 0.249 |
| InternVL-2-8B (Chen et al., 2023b) | 29.00 / 2.545 | 4.041 / 1.079 | **63.64 / 3.446** | 34.34 / **2.627** | 3.032 / 0.394 |
| AURORACAP-7B | **49.40 / 3.141** | 3.313 / 0.886 | 59.52 / 3.261 | **37.14** / 2.533 | 1.275 / 0.295 |

2015), ROUGE-L (Lin, 2004) and METEOR (Banerjee & Lavie, 2005) are included as representative rule-based metrics. For model-based metrics, we also consider Video Detailed Description (Li et al., 2023b) (VDD), which employs `ChatGPT` as an evaluation assistant to compare full captions.

Table 7 presents VDCSCORE performance across various sections of structured captions within VDC. Following (Chen et al., 2024c), we also incorporate vision-blind baselines. Furthermore, Figure 4 illustrates a schematic diagram of the performance and efficiency of different video training models. Since the comparison of model inference time under different architectures, models, deployment frameworks, and output lengths is unfair, so we used the number of visual tokens as a representation of efficiency. AURORACAP achieves superior performance in video detailed captioning while utilizing significantly fewer visual tokens than other models, fully highlighting the efficiency of AURORACAP. We also show additional experimental results with VDD metric as shown in Table 8. We also perform a human study using Elo ranking to supplement our evaluation and provide a more intuitive assessment of AURORACAP's performance. As depicted in Figure 5, VDCSCORE shows better correlation with human evaluation results than VDD and ROUGE metric.

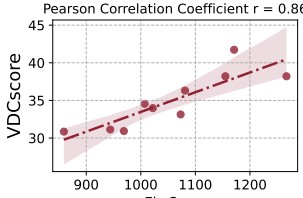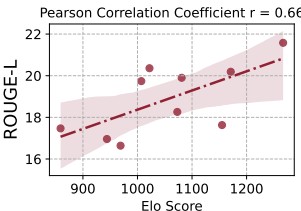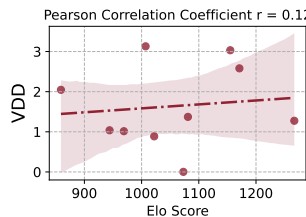

Figure 5: Pearson correlation analysis among three evaluation metrics—VDCSCORE, ROUGE-L ([Lin](Lin), 2004), VDD ([Liu et al.](Liu), 2024a)—and human Elo rankings for video models. VDD reflects the score for detailed captions. VDCSCORE demonstrates the highest consistency with expert judgments, thereby reinforcing the reliability. The detailed settings are provided in the Appendix O.

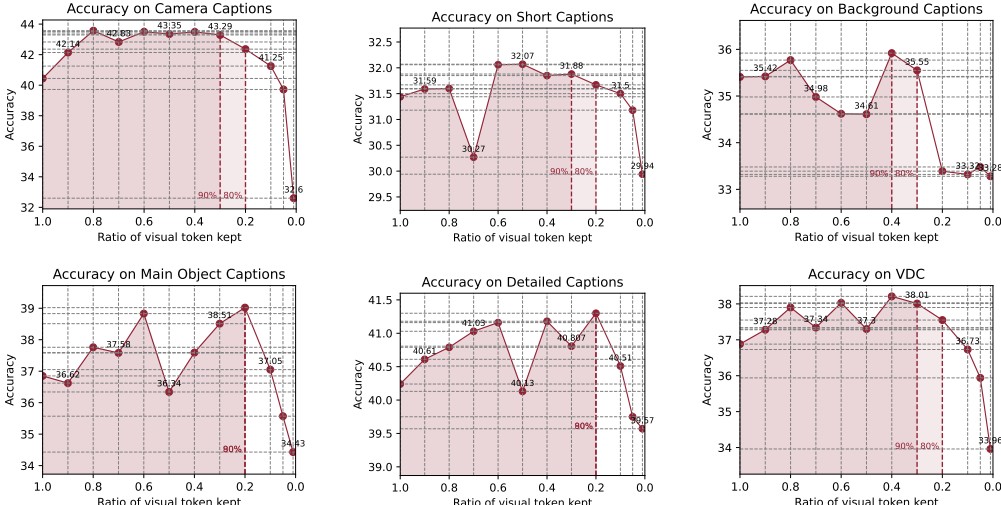

Figure 6: Ablation study of token merging on VDC. We found that token merging significantly reduces the number of tokens while maintaining minimal performance drop, and even showing improvement in some aspects. We highlight the token merging ratio when achieving 90% and 80% performance with the dash line and filled area.

**Ablation study on token merging ratio.** As a core strategy of AURORACAP, token merging plays a significant role in reducing the number of visual tokens. We further study how the video detailed captioning capability is influenced by token merge ratio. We define the performance percentage as the proportion between the highest and lowest values on the entire performance curve. As shown in Figure 6, most models maintain satisfactory performance ($> 80\%$) even with only 0.2 of visual token kept ratio. Since AURORACAP focuses on spatial visual token merging, the temporal features introduce additional complexity to explore the token merging laws, resulting in the optimal performance may occurs at a middle level of visual token kept ratio.

## 4 CONCLUSION

In this paper, we first introduce AURORACAP, a efficient video detailed captioner based on large multimodal model. By leveraging the token merging strategy, we significantly reduce the computational overhead without compromising performance. We also present VDC, a novel video detailed captioning benchmark designed to evaluate comprehensive and coherent textual descriptions of video content. For better evaluating, We propose VDCSCORE , a new LLM-assisted metric with divide-and-conquer strategy. Our extensive evaluation on various video and image captioning benchmarks demonstrated that AURORACAP achieves competitive results, even outperforming state-of-the-art models in some tasks. We also conduct thorough ablation studies to validate the effectiveness of token merging and other aspects of our model. We found that the current model performs poorly in terms of the trade-off between performance and the scale of input tokens. Additionally, there is still room for improvement in camera handling and detailed captioning.

## ACKNOWLEDGMENTS

We acknowledge Pika Labs for the computing resources support. We are also thankful to Xtuner[*], lmms-eval[*], and SGLang[*] for their well-developed and user-friendly codebase. We would like to thank Enxin Song for her contribution to the benchmark processing.

## ETHICS STATEMENT

This research on video captioning utilizes publicly available datasets, ensuring that all data complies with privacy regulations. We acknowledge the potential biases that can arise in automatic caption generation, particularly concerning gender, race, or other characteristics. We have taken measures to evaluate and minimize such biases, while remaining committed to further improvements. Additionally, we recognize the potential risks of misuse, such as generating misleading captions, and have checked the training dataset with safeguards against such applications.

## REPRODUCIBILITY STATEMENT

We have made several efforts to ensure the reproducibility of our work. All the key implementation details, including the architecture of our model, the training procedures, and hyperparameter settings, are described in supplementary meterial Section G. The introduction of the used evaluation benchmarks and settings are in Section H. The introduction of the used evaluation benchmarks and settings are in Section H. Prompt template of VDC generation is in Section K. Question-answer pairs generation prompt template of VDCSCORE is in Section L. And calculation details of Elo ranking is in Section O. We have also outlined any hardware configurations and computation requirement used for our experiments in Figure F8 to further support reproducibility. The code for model training, evaluation, deployment, as well as model weight, benchmark, and training dataset of all the training stages will be released with the paper.

---

[*]GitHub Repository: https://github.com/InternLM/xtuner
[*]GitHub Repository: https://github.com/EvolvingLMMs-Lab/lmms-eval
[*]GitHub Repository: https://github.com/sgl-project/sglang

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

# Supplementary Material

The supplementary material is structured as follows:

## A  RELATED WORKS

**Video Captioning**   The general goal of video captioning is understanding a video and describing it with natural language. Unlike image captioning, video captioning requires the description also on the temporal dimension (*e.g.,* human action, camera and object movement). The current datasets including videos in the domain of human activities (Caba Heilbron et al., 2015), cooking scene (Zhou et al., 2018; Iashin & Rahtu, 2020), movie (Song et al., 2023), and open domain (Chen & Dolan, 2011). In Table 1, we show the summary of current video captioning benchmarks. Most of them are with short caption, which is not suitable for evaluating video detailed captioning task. There are several widely used metrics to evaluate the correctness of generated caption, such as BLEU (Papineni et al., 2002), ROUGE-L (Lin, 2004), CIDEr (Vedantam et al., 2015), SPICE (Anderson et al., 2016), and BERTScore (Zhang et al., 2019). Dense Video Captioning (Yang et al., 2023a; Zhou et al., 2024) is also a captioning task but further requires localizing the events temporally. In this paper, we focus on the detailed description of a short video clip, where there are no scene changes or camera switches.

**Large Multimodal Models for Video**   With the develop of LLMs and LMMs (Maaz et al., 2023; Zhang et al., 2023b; Li et al., 2023f; Song et al., 2023; Zhang et al., 2023a; Song et al., 2024; Zhao et al., 2024a; Zhou et al., 2024; team, 2024; Jin et al., 2024; Beyer et al., 2024; Shang et al., 2024; Cai et al., 2024; Cheng et al., 2024; Li et al., 2024c), many recent works have explored adapting them into video understanding field (*e.g.,* Video-LLaMA (Li et al., 2023f), Video-LLaVA (Lin et al., 2023a), VideoChat (Li et al., 2023b), Vista-LLaMA (Ma et al., 2023), LLaVA-Hound (Zhang et al., 2024c), Koala (Tan et al., 2024), Elysium (Wang et al., 2024a), and MovieChat (Song et al., 2023; 2024)). Thanks to this flexible design, the models can combine pretrained knowledge with minimal trainable parameters. Instead of requiring large-scale training, using only a small amount of high-quality training data can even achieve better results. Most of the existing models use additional parameters for temporal modeling. There are also some interesting observations that we can actually build advancing LMMs without additional parameters for temporal modeling (Xu et al., 2024a; Lin et al., 2023b; Liu et al., 2024a; Zhang et al., 2024a; Ren et al., 2023) or even without further training with video-text data (Wu, 2024). Recent works (*e.g.,* FreeVA (Wu, 2024), LLaVA-Next (Liu et al., 2024a), VLIA (Lin et al., 2023b), and PLLaVA (Xu et al., 2024a)) also find that the vanilla LLaVA-like model pretrained on high-quality image instruction data can also be a strong video understanding model. FreeVA further observe that using existing video instruction tuning data like Video-ChatGPT 100K (Maaz et al., 2023) to tune LMMs may not necessarily lead to improvements. As the concurrent work, we

also observe this phenomenon and proceeded to develop a video detailed captioning baseline training based on the LLaVA architecture. Chat-UniVi (Jin et al., 2024) employ a set of dynamic visual tokens to uniformly represent images and videos. However, for current video inputs, when we sample at 1 FPS, the frames are already sufficiently sparse; frame-to-frame similarity is quite low except in static scenes. On the other hand, if we use keyframes as input, the frame similarity will be even lower. Under these conditions, we believe that strong temporal merging should be avoided, as it is likely to impair the model's core capabilities, especially since the CLIP vision encoder trained on images.

As for the benchmark, inspired by LLaVA (Liu et al., 2024b), Video-ChatGPT (Maaz et al., 2023) introduces a 100K video clips with text instructions with the first vLMMs benchmark evaluation system powered by LLMs. MovieChat-1K (Song et al., 2023) and CinePile (Rawal et al., 2024) are question-answering based benchmark for long-form video understanding. Shot2Story20K (Han et al., 2023) comprises videos with 2 to 8 shots each sourced for our dataset from the public video benchmark HDvila100M (Xue et al., 2022). However, currently there is no video benchmark available to evaluate video detailed captioning tasks like IIW (Garg et al., 2024) and other datasets (Li et al., 2024d) did in image captioning field. Other works (Ye et al., 2024) focus on the specific domain like Ego. In this paper, our work fills this gap. As for metric, LLaVA-Hound-DPO (Zhang et al., 2024c) uses similar divide-and-conquer strategy to build a preference dataset, ensuring that instructional data remains factually consistent with detailed captions. Our goal for VDCSCORE, however, is to evaluate the accuracy of generated captions through QA pairs generated from ground truth.

# B MORE RESULTS ON LONG-FORM VIDEO BENCHMARKS

We take MovieChat-1K (Song et al., 2023), Egoschema (Mangalam et al., 2023) as the representative long video benchmark, and compare with existing models. As shown in the following Table B2 and Table B1, AURORACAP achieves comparable performance even without training on long videos and special design. We also plot the visual token kept ratio curve on EgoSchema as Figure B1, demonstrating the stability and effectiveness of our token merging method for longer video sequences.

Table B1: Model performance on Egoschema (Mangalam et al., 2023) benchmark.

| Model | Design For Long Video | Accuracy |
|---|:---:|:---:|
| Random Choice | - | 20.0 |
| FrozenBiLM | × | 26.9 |
| mPLUG-Owl | × | 31.1 |
| TimeChat | ✓ | 33.0 |
| Video-LLAVA | × | 38.4 |
| LLAMA-VID | × | 38.5 |
| LLAVA-NeXT-Video | × | 43.9 |
| VideoLLAMA2 | × | 51.7 |
| MovieChat | ✓ | 53.5 |
| Human Performance | - | 76.2 |
| AURORACAP (ours) | × | 46.0 |

Table B2: Model performance on MovieChat-1K (Song et al., 2023) benchmark.

| Model | Degign for Long Video | Breakpoint | Global |
|---|:---:|:---:|:---:|
| Video-LLaMA | × | 39.1 | 51.7 |
| VideoChat | × | 46.1 | 57.8 |
| TimeChat | ✓ | 46.1 | 73.8 |
| VideoChatGPT | × | 48.0 | 47.6 |
| MovieChat | ✓ | 48.3 | 62.3 |
| MovieChat+ | ✓ | 49.6 | 71.2 |
| AURORACAP (ours) | × | 52.6 | 59.7 |

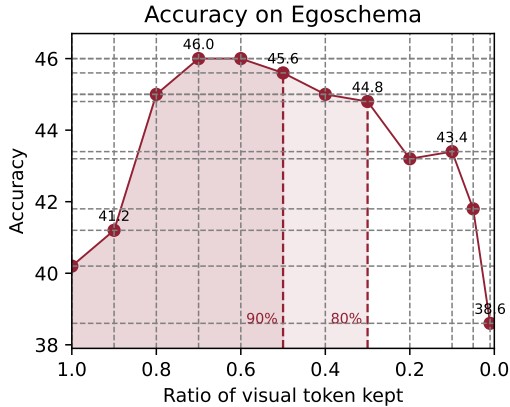

Figure B1: Token merging ablation curve on Egoschema (Mangalam et al., 2023). The best performance is set range from 0.6 to 0.7 of visual token kept ratio.

## C  N-GRAM BASED EVALUATION RESULTS

A variant of VDCSCORE could be developed to assess final triplets (`<question, correct answer, predicted answer>`) by employing a n-gram based evaluation approach as shown in Table C3 with correlation shown in Figure C2.

Table C3: Evaluation results of various models on VDC based on ROUGE for answer matching.

| Model | Avg. | Camera | Short | Background | Main Object | Detailed |
|---|---|---|---|---|---|---|
| LLaMA-VID | 6.66 | 9.53 | 5.12 | 5.30 | 6.07 | 7.29 |
| Video-ChatGPT-7B | 6.85 | 9.20 | 5.37 | 6.40 | 6.29 | 6.97 |
| ViLA-7B | 6.68 | 7.88 | 5.16 | 6.29 | 6.35 | 7.74 |
| Video-LLAVA-7B | 6.79 | 8.67 | 5.25 | 6.09 | 6.64 | 7.28 |
| LLAVA-1.5-7B | 6.34 | 8.31 | 5.40 | 5.80 | 5.52 | 6.65 |
| LongVA-7B | 6.70 | 8.37 | 4.99 | 6.11 | 6.79 | 7.23 |
| LLAVA-1.5-13B | 6.34 | 8.64 | 5.55 | 5.57 | 5.15 | 6.79 |
| LLAVA-NeXT-V7B | 6.69 | 8.61 | 5.30 | 6.35 | 6.27 | 6.93 |
| LLAVA-1.6-7B | 6.45 | 8.08 | 5.38 | 5.79 | 5.82 | 7.20 |
| LLAVA-1.6-13B | 6.18 | 7.31 | 5.28 | 6.00 | 5.29 | 7.01 |
| ShareGPT4Video-8B | 6.78 | 8.46 | 5.73 | 5.82 | 6.54 | 7.36 |
| LLAVA-OV-7B | 6.59 | 8.09 | 5.66 | 5.86 | 5.58 | 7.75 |
| InternVL-2-8B | 6.82 | 8.14 | 5.63 | 5.95 | 6.53 | 7.86 |
| AURORACAP-7B | 7.60 | 9.55 | 5.66 | 6.79 | 7.42 | 8.58 |
| Gemini-1.5 Pro | 7.59 | 8.78 | 6.25 | 7.29 | 7.70 | 7.94 |

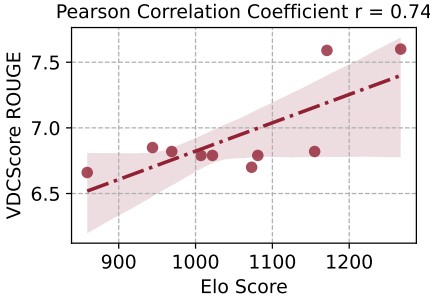

Figure C2: Correlation between the n-gram evaluation scores and human elo ranking on the VDC.

# D    BENCHMARK COMPARISON

We compare our proposed VDCwith some examples from several video captioning benchmarks as shown in Figure D3, Figure D4, and Figure D5. The corresponding captions are shown as followings:

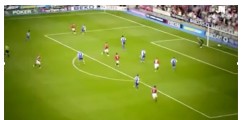 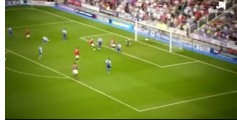 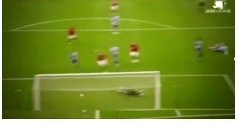 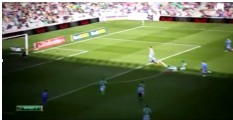

Figure D3: Video example of MSR-VTT (Xu et al., 2016) benchmark.

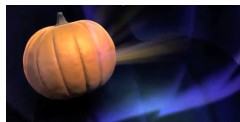 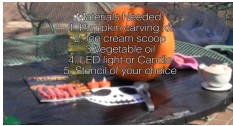 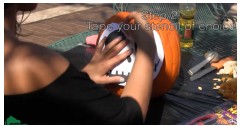 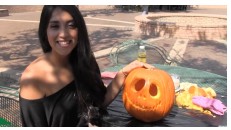

Figure D4: Video example of VATEX (Wang et al., 2019) benchmark.

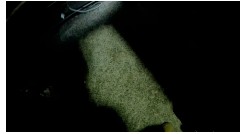 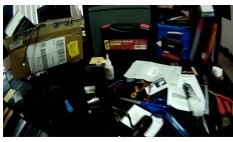 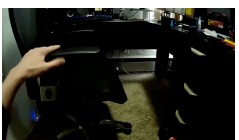 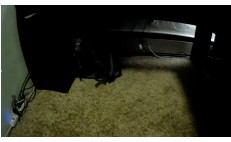

Figure D5: Video example of VDC benchmark sampled from Ego4D (Grauman et al., 2022).

**Benchmark   Caption (for VDC we show only the detailed caption)**

MSR-VTT D3    (4 words) Teams are playing soccer.

VATEX D4    (13 words) A woman instructs and demonstrates how to remove the insides of a pumpkin.

VDC D5    (618 words) The video opens with an intimate close-up of a surface adorned with vibrant green moss and intricate lichen, initially evoking the serene beauty of a natural landscape. This organic imagery quickly transitions, revealing that the mossy surface is actually part of a motorcycle or vehicle's engine compartment, creating a striking contrast between the lush textures of nature and the cold, hard lines of mechanical components. As the camera angle shifts, the viewer is drawn deeper into the engine compartment, where the interplay of moss overgrowth on various machinery introduces a fascinating blend of organic life and industrial elements, highlighting the unexpected coexistence of nature and technology.

The perspective then zooms in, accentuating the rich details of the mossy growth, which clings tenaciously to the metallic surfaces, while a dark cavity beneath hints at the complexity of the machinery. The reflective metallic surfaces glint in the light, further enhancing the visual contrast and inviting the viewer to explore this unique juxtaposition. Suddenly, the scene shifts dramatically, with rapid camera motion creating a vibrant blur of colors and shapes, transforming the previously detailed views into a chaotic whirlwind, suggesting a swift movement through the intricate landscape of the engine compartment.

As the motion blur begins to dissipate, the viewer is presented with a clearer image of a light-colored, textured surface, where blurred mechanical components can be discerned, indicating a deceleration in movement. The camera stabilizes, revealing a rough-textured floor or ground that suggests an indoor or industrial environment, characterized by a sense of organized chaos. The scene transitions to a detailed examination of a cluttered workspace filled with tangled wires, casings, and components in a variety of colors, emphasizing the disorganized state of electronic or mechanical internals, possibly during a maintenance or repair process.

The perspective shifts once more, showcasing darker, textured surfaces juxtaposed against lighter insulating materials, with hidden metallic elements peeking through, suggesting another angle within this same cluttered interior space. A human hand enters the frame, reaching out to interact with the components, signaling an active workspace filled with purpose. As the scene expands, additional hands join the fray, actively manipulating various objects within the crowded environment, signifying an ongoing task or collaborative effort amidst the complex array of components and materials.

The atmosphere is imbued with a sense of urgency and engagement, as the camera captures the dynamic interactions of the individuals working together. The camera work remains fluid and dynamic, featuring a mix of close-up shots that highlight the intricate details of the components and wider angles that provide context to the bustling environment. The slightly shaky nature of the shots adds a layer of realism and immersion, drawing the viewer into the heart of the action. The low light conditions create a moody ambiance, with shadows dancing across the surfaces, enhancing the visual depth and interest of the scene. Overall, the video encapsulates a vivid portrayal of the intersection between nature and machinery, as well as the collaborative spirit of those engaged in the intricate task of maintenance and repair within this unique setting.

## E  TOKEN MERGING VISUALIZATION

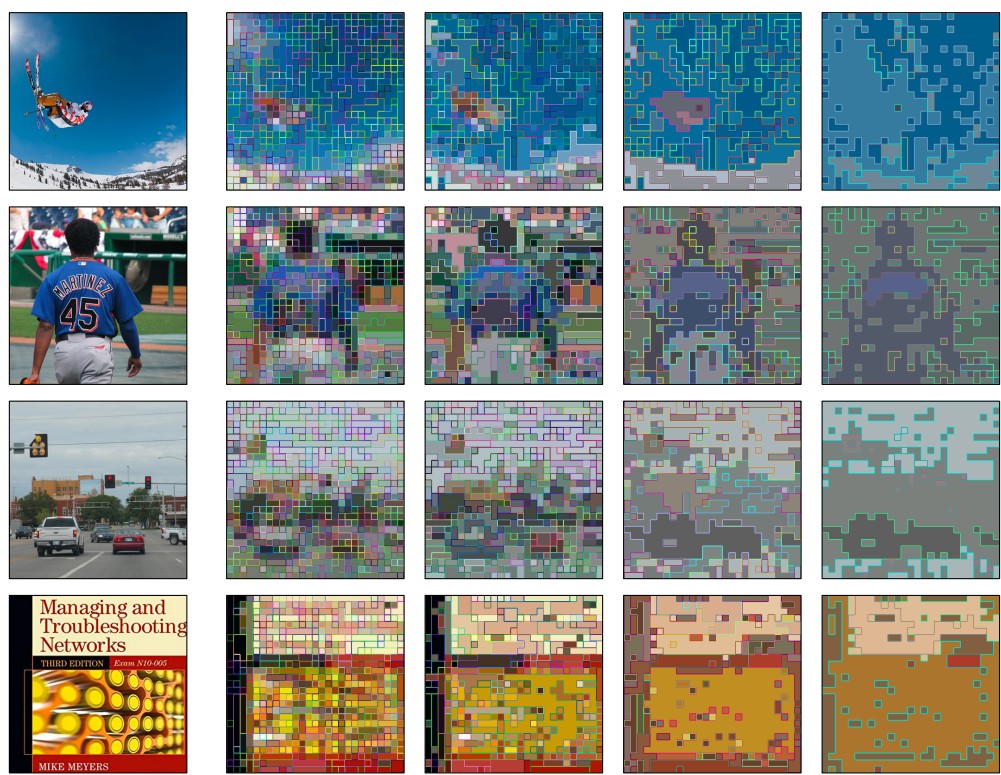

Figure E6: Token merging visualization. From top to bottom, the image IDs are COCO:COCO-train2014-000000247906, VG:2331508, SA-1B:sa-393200, OCR-VQA:71789111. From left to right, the number of tokens representing the images are 490, 154, 18, and 6.

The number of tokens $r$ merged in each layer of the transformer in ViT is calculated by the following formula:

$$r = (1 - \text{ratio}) \cdot (W \cdot H / P^2) / L \tag{2}$$

where $W$, $H$ is the width and the height of the input image, $P$ is the patch size, $L$ is the number of layers in the transformer, ratio is the proportion of tokens to be kept. The merging is evenly distributed across all layers of the transformer.

We show token merging visualization examples from COCO (Lin et al., 2014), VG (Krishna et al., 2017b), SA-1B (Kirillov et al., 2023), and OCR-VQA (Mishra et al., 2019) in Figure E6, since we only conduct frame-wise operation. Features extracted by CLIP ViT-H model [*]. We conduct image captioning for the examples given final token numbers by 7B AURORACAP as following. We remove line breaks in generated caption for a better reading experience.

---

[*]Huggingface Model: https://huggingface.co/apple/DFN5B-CLIP-ViT-H-14-378

| # Token | **Describe this image in detail. (Figure E6 Row 1)** |
|---|---|
| 490 | In the heart of a winter wonderland, a skier takes flight, suspended in mid-air against the backdrop of a clear blue sky. The skier, clad in a vibrant red and white jacket, is captured in a moment of pure exhilaration. The skier's body is angled towards the left side of the image, with their skis crossed in an impressive display of skill and control. The skier's helmet, a stark contrast to the surrounding snow, is adorned with a pink and white design, adding a touch of color to the otherwise monochrome landscape. The skier's goggles, reflecting the bright sunlight, are a testament to the intensity of the sport. Beneath the skier, the snow-covered mountain slopes stretch out, dotted with trees that stand as silent spectators to the skier's daring feat. The image is a freeze-frame of action and adventure, encapsulating the thrill and beauty of winter sports. |
| 154 | In the midst of a clear blue sky, a skier is captured in a moment of thrilling action. The skier, clad in a vibrant red and white jacket, is suspended in mid-air, performing an impressive trick. The skier's body is parallel to the ground, creating a sense of motion and excitement. The skier's skis, a striking combination of red and white, are crossed in front of them, adding to the dynamic nature of the image. The skier's helmet, a necessary safety measure, is white and contrasts with the colorful jacket. The backdrop to this action-packed scene is a breathtaking snowy mountain landscape. The mountain's peak is visible in the distance, adding depth to the image. The overall composition of the image suggests a high-speed descent down the mountain, with the skier skillfully navigating the terrain. This image encapsulates the exhilarating sport of skiing, showcasing not only the skill of the skier but also the stunning beauty of the mountain landscape. |
| 18 | In the heart of a snowy mountain, a daring skier takes flight. Dressed in a vibrant red and white jacket, black pants, and a protective white helmet, the skier is captured mid-air, performing an impressive trick. The skier's body is parallel to the ground, arms outstretched, as if embracing the thrill of the moment. The skier's skis, a striking combination of red and white, are pointed upwards towards the clear blue sky. The skier's poles, held firmly in their hands, add to the dynamic nature of the scene. The backdrop to this exhilarating moment is a breathtaking snowy mountain landscape. The mountain's peak, blanketed in snow, stretches out into the distance. The clear blue sky above is devoid of clouds, adding to the serene beauty of the scene. This image captures not just a moment, but a story of adventure, thrill, and the sheer joy of skiing. |
| 6 | The image captures a thrilling moment of a snowboarder in mid-air, performing an impressive trick. The snowboarder, clad in a vibrant red and white jacket, black pants, and a protective helmet, is the central figure in this dynamic scene. The snowboarder is holding onto a rope with one hand, suggesting that they are being pulled up the mountain by a snowmobile, a common practice in snowboarding to gain speed and momentum. The rope extends from the top left corner of the image, creating a sense of movement and direction. The backdrop of the image is a breathtaking snowy mountain landscape, complete with trees and other natural elements. The snowboarder is positioned in the center of the image, drawing the viewer's attention to their daring stunt. The image is a testament to the exhilarating sport of snowboarding and the skill and courage of the athletes who participate in it. It's a freeze-frame of an adrenaline-fueled moment, encapsulating the thrill and excitement of snowboarding. |

| # Token | **What is the number in the picture? (Figure E6 Row 2)** |
|---|---|
| 490 | The number in the picture is "45". It is displayed on the back of the jersey that the person is wearing. |
| 154 | The number in the picture is "45". |
| 18 | The number in the picture is "45". It is displayed on the back of the baseball jersey worn by the player in the image. |
| 6 | The number in the picture is "45". It is displayed in a large, bold font on the jersey of the baseball player in the image. |

| # Token | **What is the relative position of the white and red cars in the picture? Which one in the left? (Figure E6 Row 3)** |
|---|---|
| 490 | The white car is positioned to the left of the red car in the image. The white car is closer to the foreground and is stopped at the traffic light, while the red car is further back, also stopped at the same traffic light. The relative position of the cars is such that the white car is in the left lane and the red car is in the right lane, both facing the same direction of traffic. |
| 154 | The white car is positioned to the left of the red car in the image. The red car is further back on the road, indicating it is behind the white car. This is determined by the perspective of the image, where the white car is closer to the foreground and the red car is further away, both in the same lane and moving in the same direction. |
| 18 | The white car is on the left side of the image, and the red car is on the right side. The white car is closer to the foreground, while the red car is further away, positioned behind the white car. |
| 6 | The white car is on the left side of the image, while the red car is on the right side. The white car is positioned closer to the foreground, while the red car is further back, near the intersection. |

## F ABLATION STUDIES

In this section, we conduct the ablation studies in several aspects including token merging, slowfast inference, efficiency, training strategy, and backbone selection.

**Token merging.** As a core strategy of AURORACAP, token merging plays a significant role in reducing the number of visual tokens. We conduct extensive ablation studies to explore the impact of the token kept ratio $R_{vtk}$ in terms of performance across multiple tasks including image captioning, visual question answering, video captioning, and video question answering as shown in Figure F9 and Figure F10. We define the performance percentage as the proportion between the highest and lowest values on the entire performance curve. We identify the minimum retention thresholds for achieving 90% and 80% performance. As shown in Figure F9, while the performance of AURORACAP generally declines with fewer visual tokens across most benchmarks, it remains relatively stable at higher retention levels. Most models maintain satisfactory performance ($> 80\%$) even with only 0.4 of $R_{vtk}$, highlighting the efficiency of our approach. Visual token retention thresholds vary by task complexity, with more visually demanding tasks needing higher retention of visual tokens. For instance, CIDEr (Vedantam et al., 2015) on COCO-Cap (Veit et al., 2016) maintains over 90% performance with an $R_{vtk}$ of 0.3, whereas accuracy on GQA (Hudson & Manning, 2019) drops to 90% when the $R_{vtk}$ is reduced to 0.8. Unlike image understanding, the optimal performance across most video understanding benchmarks occurs at a relatively low $R_{vtk}$ as depicted in Figure F10. And for MSR-VTT (Xu et al., 2016), VATEX (Wang et al., 2019), and ActivityNet-QA (Yu et al., 2019), even achieve better results at extremely low $R_{vtk}$ ($< 0.1$). It indicates that comparing to image, video input have higher redundancy. Note that AURORACAP focuses on spatial visual token merging, while the temporal features introduce additional complexity to explore the token merging laws. Appendix E shows more calculation details and the visualization results of token merging.

**Slowfast inference.** Inspired by Slowfast-LLaVA (Xu et al., 2024b), we explore whether combining frames with low and high $R_{vtk}$ can enhance performance. In practice, we don't conduct token merging in the first frame and concatenate them with the merged tokens from subsequent frames. We apply this strategy to both video captioning and video question answering tasks, comparing performance with and without the inclusion of full first-frame visual tokens. As illustrated in Table F4, slowfast inference brings marginal performance improvement in video question answering tasks or even drop in video captioning tasks but with more computing cost. Therefore, by default, we don't using slowfast inference for video detailed captioning.

We also present the performance curve with and without the inclusion of full first-frame visual tokens as the visual token kept ratio varies during inference across multiple video understanding tasks. As illustrated in Figure F11 and Figure F12, despite the inclusion of full first-frame visual tokens, slowfast inference does not consistently result in significantly positive effects on performance. In some cases, the incorporation of full first-frame visual tokens even worsens performance degradation as the kept ratio decreases, particularly in video captioning tasks.

Table F4: Ablation on slowfast inference for AURORACAP-7B. We present the average performance among different token merging ratio on various video understanding benchmarks. We show that slowfast inference brings marginal performance improvement in video question answering tasks or even drop in video captioning tasks but with more computing cost.

| Setting | MSR-VTT | | | | | VATEX | | | | | ANet | MSVD | MSRVTT |
|---|---|---|---|---|---|---|---|---|---|---|---|---|---|
| | $\overline{C}$ | $\overline{B@1}$ | $\overline{B@4}$ | $\overline{M}$ | $\overline{R}$ | $\overline{C}$ | $\overline{B@1}$ | $\overline{B@4}$ | $\overline{M}$ | $\overline{R}$ | $\overline{Acc}$ | $\overline{Acc}$ | $\overline{Acc}$ |
| w/o slowfast | 26.72 | 53.01 | 17.58 | 21.25 | 46.78 | 28.03 | 52.28 | 15.22 | 16.95 | 38.30 | 58.55 | 56.45 | 37.26 |
| w/ slowfast | 26.18 | 51.68 | 17.00 | 21.20 | 46.16 | 28.07 | 52.16 | 15.12 | 16.95 | 38.27 | 59.66 | 55.65 | 38.22 |
| Δ | -0.54 | -1.33 | -0.58 | -0.05 | -0.62 | +0.04 | -0.12 | -0.10 | -0.01 | -0.03 | +1.11 | -0.80 | +0.96 |

**Efficiency.** To assess the inference speed, we utilize the inference time per video question-answering pair in seconds (TPV) as an evaluative metric. SGLang is an accelerated serving framework for LLMs and multimodal LLMs. We consider four settings including with or without token merging and SGLang. Figure F7 indicates the minimum TPV achievable in each settings across seven video understanding datasets. Reducing the visual tokens and using SGLang result in excellent inference times per video question-answering pair while all the datasets with short video and question inputs.

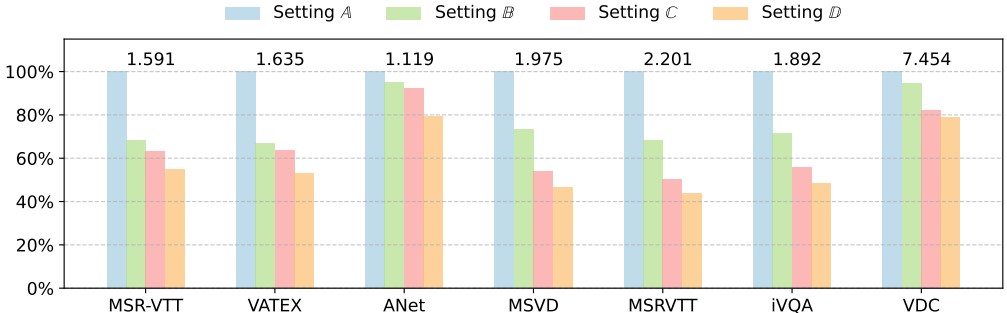

Figure F7: Comparison between different inference settings: $\mathbb{A}$: $R_{vtk} = 1.0$, without SGLang, $\mathbb{B}$: $R_{vtk} = 0.1$, without SGLang, $\mathbb{C}$: $R_{vtk} = 1.0$, with SGLang, $\mathbb{D}$: $R_{vtk} = 0.1$, with SGLang. The number indicates the maximum inference time in seconds for each benchmark.

In contrast, maintaining full visual tokens or omitting the use of SGLang results in comparatively slower performance, demonstrating the superior inference efficiency of AURORACAP. For each input, we process the video at a resolution of $378 \times 378$ and sample 8 frames using single H100 GPU.

**Training strategy.** Alternative training strategies for the language stage of AURORACAP are less frequently explored, which is the primary focus of this section. For a fair comparison, we use the same training datasets across all settings and maintain consistent hyper-parameters. The following training settings are explored:

- **Setting $\mathbb{A}$:** End-to-end training and set $R_{vtk}$ to 1.0.
- **Setting $\mathbb{B}$:** Following Slowfast-LLaVA (Xu et al., 2024b), we retain full visual tokens in the first frame and concatenate with merged tokens using $R_{vtk}$ of 0.1.
- **Setting $\mathbb{C}$:** End-to-end training and set $R_{vtk}$ to 0.1.
- **Setting $\mathbb{D}$:** Most videos in the training data have no more than 8 key frames, while a subset (mainly from ShareGPT4Video (Chen et al., 2024d)) contains significantly more. We first exclude this subset from end-to-end training and then use it to train solely the LLM, enhancing its ability to handle multi-frame inputs.
- **Setting $\mathbb{E}$:** Training solely the LLM match the performance of Setting $\mathbb{A}$ set $R_{vtk}$ at 0.1.

We implement these training strategies, track training costs in H100 hours, and evaluate across various video understanding tasks. As shown in Figure F8, while training with an $R_{vtk}$ of 1.0 improves performance, it significantly increases training time. Surprisingly, mixing lower and higher visual token ratios during training offers no significant advantage. Training only the LLM under the two settings results in a performance drop, indicating that enhancing long video understanding still requires collaboration with the finetuning the visual encoder. Therefore, we choose Setting $\mathbb{C}$ as the final training strategy.

**Backbone selection.** We use the the training loss among the last ten iterations in original LLaVA alignment pretraining stage to guidance the ViT and LLM backbones selection as shown in Table F5.

## G    DETAILED TRAINING SETTINGS

We use CLIP ViT-H [*], Vicuna-1.5-7B [*] as the initialization of AURORACAP-7B. Training hyper-parameters for both stages are shown in Table G9. For visual data preprocessing, we resize each image or video frame to the short side of 378 while keeping original aspect. Instead of doing center crop, we conduct bilinear position embedding interpolation for each training sample. For video data, we extract frames at 2 FPS uniformly. For token merging, we use constant schedule for each blocks.

---

[*]Huggingface Model: https://huggingface.co/apple/DFN5B-CLIP-ViT-H-14-378
[*]Huggingface Model: https://huggingface.co/lmsys/vicuna-7b-v1.5-16k

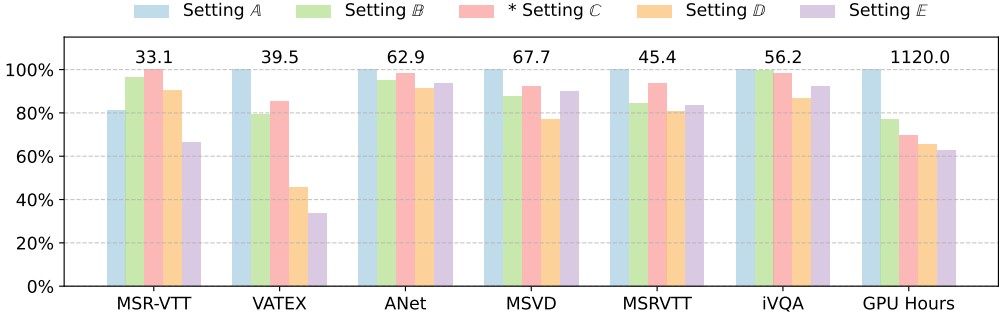

Figure F8: Comparison between different training strategy in Language stage. We take Accuracy for Question-Answering tasks and CIDEr for captioning tasks as the evaluation metric and present the performance percentage. We choose Setting ℂ as the final training strategy as shown with *. The number shows the maximum value for each benchmark.

Table F5: Final training loss during pretraining stage with original LLaVA pretraining data.

| ViT | ViT Size | LLM | LLM Size | Loss |
|---|---|---|---|---|
| facebook/dinov2-giant | 1,136M | microsoft/phi-2 | 2.7B | 3.3021 |
| openai/clip-vit-large-patch14-336 | 428M | Qwen/Qwen1.5-0.5B-Chat | 0.5B | 3.1001 |
| openai/clip-vit-large-patch14-336 | 428M | microsoft/phi-2 | 2.7B | 2.8067 |
| laion/CLIP-ViT-bigG-14-laion2B-39B-b160k | 1,845M | microsoft/phi-2 | 2.7B | 2.7124 |
| facebook/dinov2-giant | 1,136M | lmsys/vicuna-13b-v1.5 | 13B | 2.3895 |
| laion/CLIP-ViT-bigG-14-laion2B-39B-b160k | 1,845M | internlm/internlm2-chat-7b | 7B | 2.3437 |
| laion/CLIP-ViT-bigG-14-laion2B-39B-b160k | 1,845M | internlm/internlm2-chat-20b | 20B | 2.2745 |
| laion/CLIP-ViT-bigG-14-laion2B-39B-b160k | 1,845M | deepseek-ai/deepseek-llm-67b-chat | 67B | 2.1572 |
| laion/CLIP-ViT-bigG-14-laion2B-39B-b160k | 1,845M | mistralai/Mistral-7B-Instruct-v0.1 | 7B | 2.1569 |
| openai/clip-vit-large-patch14-336 | 428M | mistralai/Mixtral-8x7B-Instruct-v0.1 | 8x7B | 2.0815 |
| apple/DFN5B-CLIP-ViT-H-14-378 | 632M | lmsys/vicuna-13b-v1.5-16k | 13B | 2.0443 |
| laion/CLIP-ViT-bigG-14-laion2B-39B-b160k | 1,845M | lmsys/vicuna-7b-v1.5-16k | 7B | 2.0365 |
| laion/CLIP-ViT-bigG-14-laion2B-39B-b160k | 1,845M | mistralai/Mixtral-8x7B-Instruct-v0.1 | 8x7B | 1.9889 |
| openai/clip-vit-large-patch14-336 | 428M | lmsys/vicuna-7b-v1.5 | 7B | 1.9762 |
| laion/CLIP-ViT-bigG-14-laion2B-39B-b160k | 1,845M | meta-llama/Llama-2-13b-chat-hf | 13B | 1.9708 |
| laion/CLIP-ViT-bigG-14-laion2B-39B-b160k | 1,845M | lmsys/vicuna-13b-v1.5 | 13B | 1.9412 |
| apple/DFN5B-CLIP-ViT-H-14-378 | 632M | lmsys/vicuna-7b-v1.5-16k | 7B | **1.8679** |

# H    EVALUATION BENCHMARKS AND SETTINGS

We list all the hyper-parameters and prompt used for evaluation as shown in Table H10. For our proposed VDC, we show the settings as following with the max number of the tokens of 1,024:

| Type | Prompt used for evaluation |
|---|---|
| Camera | Describe any camera zooms, pans, or angle changes. |
| Background | Summarize the background setting of the video based on these frames. |
| Main Object | Describe the main subject, including their attributes and movements throughout the video. |
| Detail | Imagine the video from these frames and describe it in detail. |

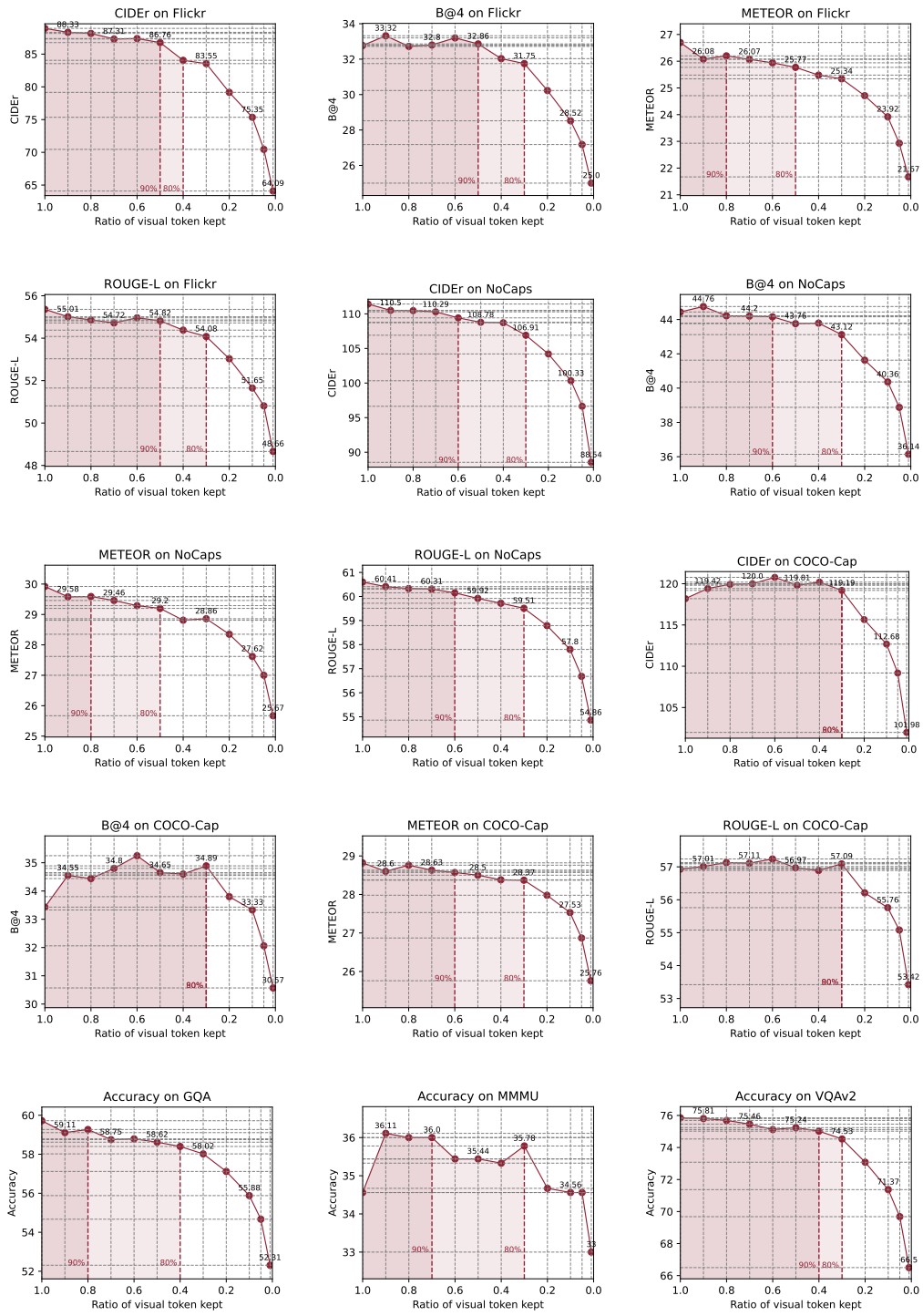

Figure F9: Ablation study of token merging on image captioning on Flickr (Young et al., 2014), NoCaps (Agrawal et al., 2019), COCO-Cap (Lin et al., 2014), visual question answering in GQA (Hudson & Manning, 2019), MMMU (Yue et al., 2023), VQAv2 (Goyal et al., 2017). We found that token merging significantly reduces the number of tokens while maintaining minimal performance drop, and even showing improvement in some tasks. We highlight the token merging ratio when achieving 90% and 80% performance with the dash line and filled area.

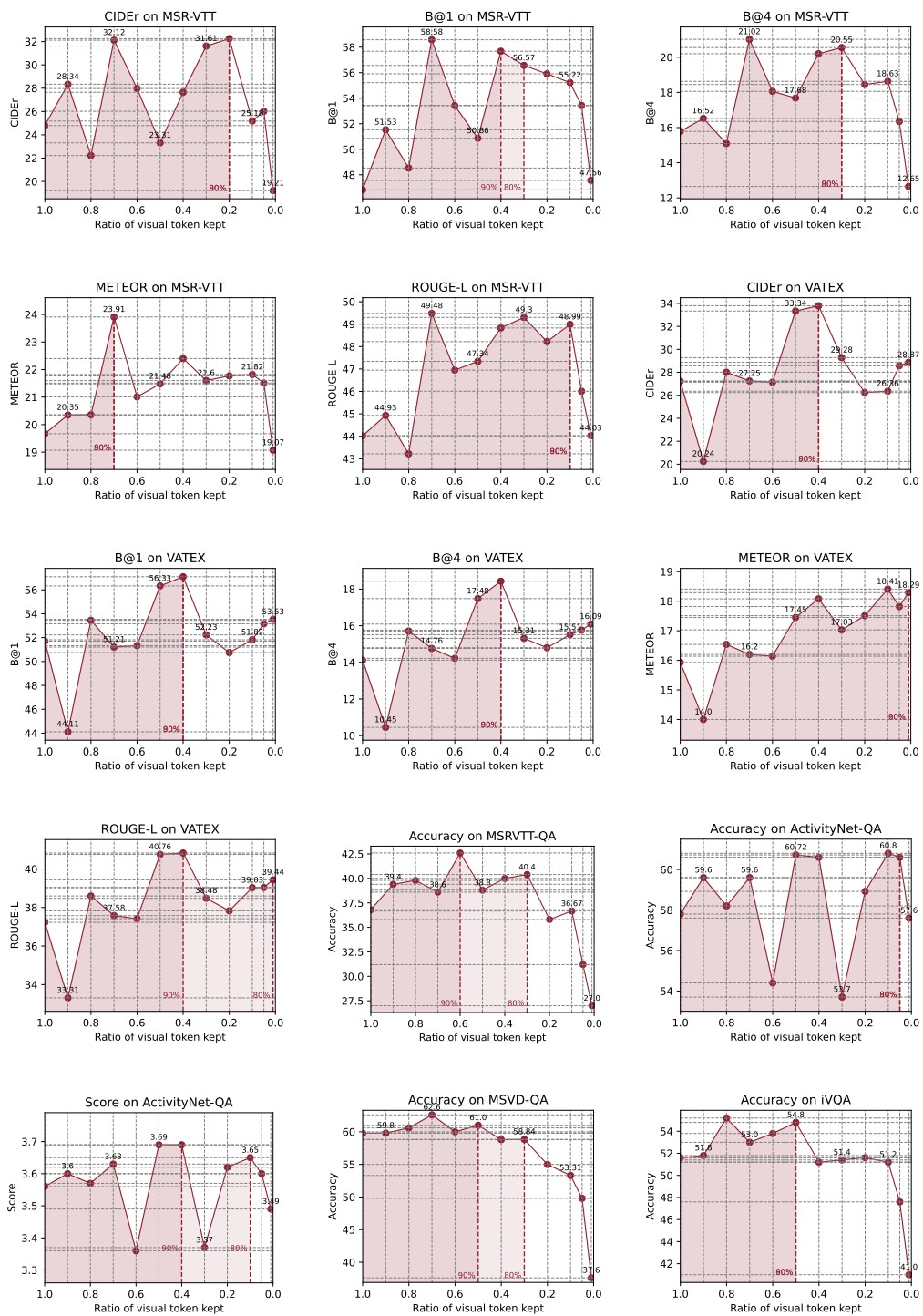

Figure F10: Ablation study of token merging on video captioning on MSRVTT (Xu et al., 2016), VATEX (Wang et al., 2019), video question answering on MSRVTT-QA (Xu et al., 2016), ActivityNet-QA (Yu et al., 2019), MSVD-QA (Xu et al., 2017), iVQA (Liu et al., 2018). We found that token merging significantly reduces the number of tokens while maintaining minimal performance drop, and even showing improvement in some tasks. We highlight the token merging ratio when achieving 90% and 80% performance with the dash line and filled area.

Table G6: Summary of datasets used for training AURORACAP in Pretraining stage.

| Task | # Sample | Dataset |
|---|---|---|
| Image Captioning | 1.3M | LAION-CC-SBU-595K (Liu et al., 2024b), ShareGPT4V (Chen et al., 2023a), ALLaVA-Caption-LAION-4V (Chen et al., 2024a), ALLaVA-Caption-VFLAN-4V (Chen et al., 2024a), DenseFusion (Li et al., 2024e) |

Table G7: Summary of datasets used for training AURORACAP in Vision stage. For classification, Reasoning, VQA, and Generation tasks, we adopt the dataset processed by M³IT (Li et al., 2023c) to fit the training objective of language models.

| Task | # Sample | Dataset |
|---|---|---|
| Captioning | 1,925K | ShareGPT4V-PT (Chen et al., 2023a), TextCaps (Sidorov et al., 2020), Image-Paragraph-Captioning (Krause et al., 2017) |
| Object-centric | 438K | COST (Jain et al., 2023), ChatterBox (Tian et al., 2024), V* (Wu & Xie, 2023) |
| Classification | 238K | COCO-GOI (Lin et al., 2014), COCO-Text (Veit et al., 2016), ImageNet (Russakovsky et al., 2015), COCO-ITM (Lin et al., 2014), e-SNLI-VE (Kayser et al., 2021), Mocheg (Yao et al., 2023), IQA (Duanmu et al., 2021) |
| Reasoning | 100K | CLEVR (Johnson et al., 2017), NLVR (Suhr et al., 2017), VCR (Zellers et al., 2019), VisualMRC (Tanaka et al., 2021), Winoground (Thrush et al., 2022) |
| VQA | 3,518K | VQA v2 (Goyal et al., 2017), Shapes VQA (Andreas et al., 2016), DocVQA (Mathew et al., 2021), OK-VQA (Marino et al., 2019), Text-VQA (Singh et al., 2019), OCR-VQA (Mishra et al., 2019), A-OK-VQA (Schwenk et al., 2022), ScienceQA (Lu et al., 2022), ST-VQA (Biten et al., 2019), ViQuAE (Lerner et al., 2022), LLaVA-OneVision (Li et al., 2024a) |
| Generation | 145K | Visual Storytelling (Huang et al., 2016), Visual Dialog (Das et al., 2017), Multi30k (Elliott et al., 2016) |
| Chinese | 193K | COCO-Caption CN (Li et al., 2019), Flickr-8k-Caption CN (Li et al., 2016), multimodal Chat (Zheng et al., 2021), FM-IQA (Gao et al., 2015), ChineseFoodNet (Chen et al., 2017) |
| Total | 6.6M | For all datasets, we uniformly sample without duplication. |

Table G8: Summary of datasets used for training AURORACAP in Language stage.

| Task | # Sample | Dataset |
|---|---|---|
| Image Captioning | 1,779K | ShareGPT4V (Chen et al., 2023a), ALLaVA-Caption-LAION-4V (Chen et al., 2024a), ALLaVA-Caption-VFLAN-4V (Chen et al., 2024a), DenseFusion (Li et al., 2024e), FaceCaption (Dai et al., 2024a) |
| Video Captionin | 1,659K | MiraData (Ju et al., 2024), LLaVA-Hound (Zhang et al., 2024c), ShareGPT4Video (Chen et al., 2024d), Private Data |
| Image Instruction | 9,742K | LLaVA-Mix-665K (Liu et al., 2023), LVIS-Instruct4V (Wang et al., 2023), ALLaVA-Instruct-LAION-4V (Chen et al., 2024a), ALLaVA-Instruct-VFLAN-4V (Chen et al., 2024a), Cambrian (Tong et al., 2024a), M4-Instruct (Liu et al., 2024a) |
| Video Instruction | 268k | LLaVA-Hound (Zhang et al., 2024c), ShareGPT4Video (Chen et al., 2024d) |
| Language-only | 143K | Evol-Intruct-GPT4-Turbo-143K (Chen et al., 2024a) |
| Total | 15.0M | We duplicate video captioning and instruction datasets twice. |

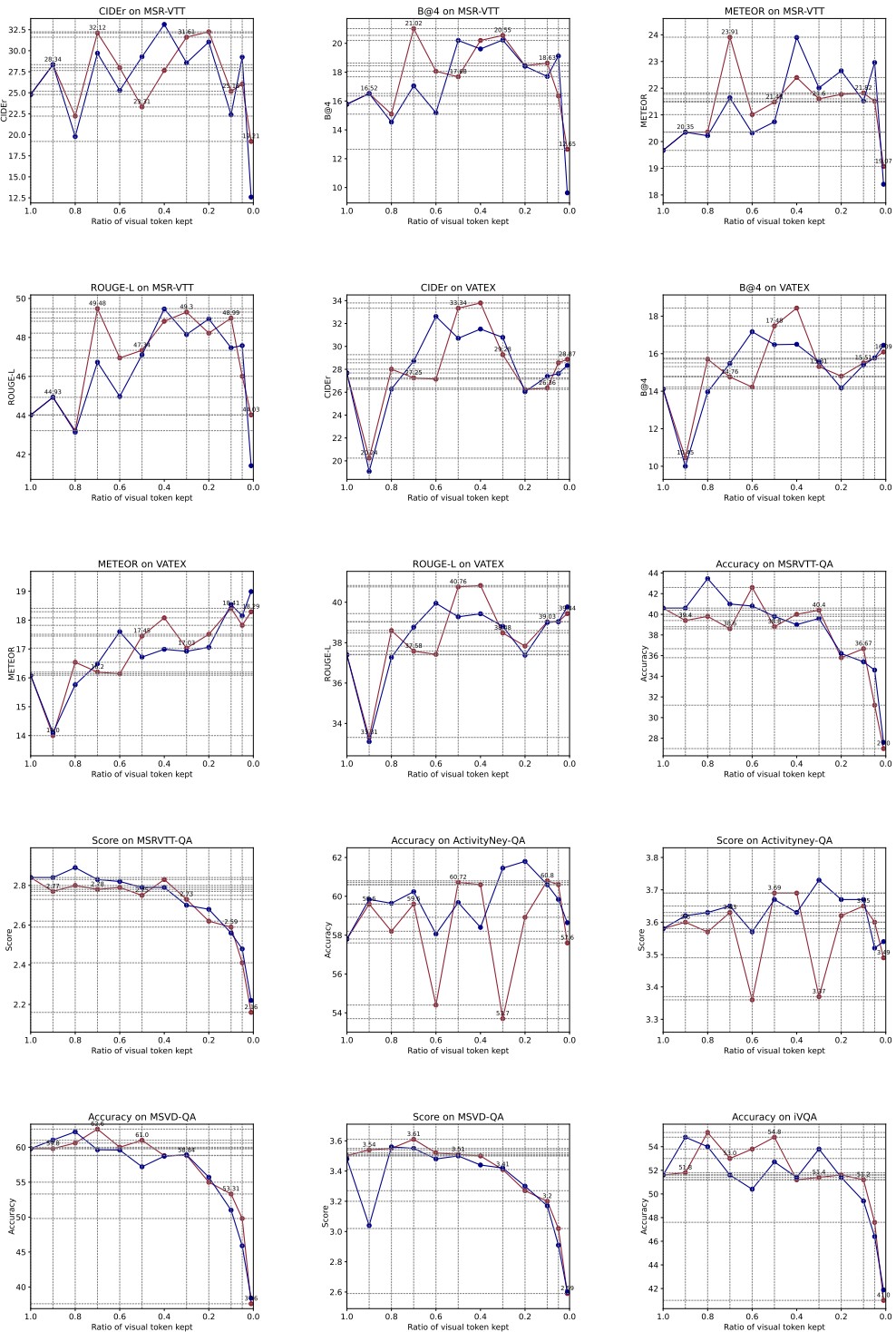

Figure F11: Comparison between performance with and without the inclusion of full first-frame visual tokens during inference on video captioning on MSRVTT (Xu et al., 2016), VATEX (Wang et al., 2019), video question answering on MSRVTT-QA (Xu et al., 2016), ActivityNet-QA (Yu et al., 2019), MSVD-QA (Xu et al., 2017), iVQA (Liu et al., 2018). As the visual token kept ratio varies, the blue curve indicates performance with slowfast inference, while the red curve represents performance without slowfast inference.

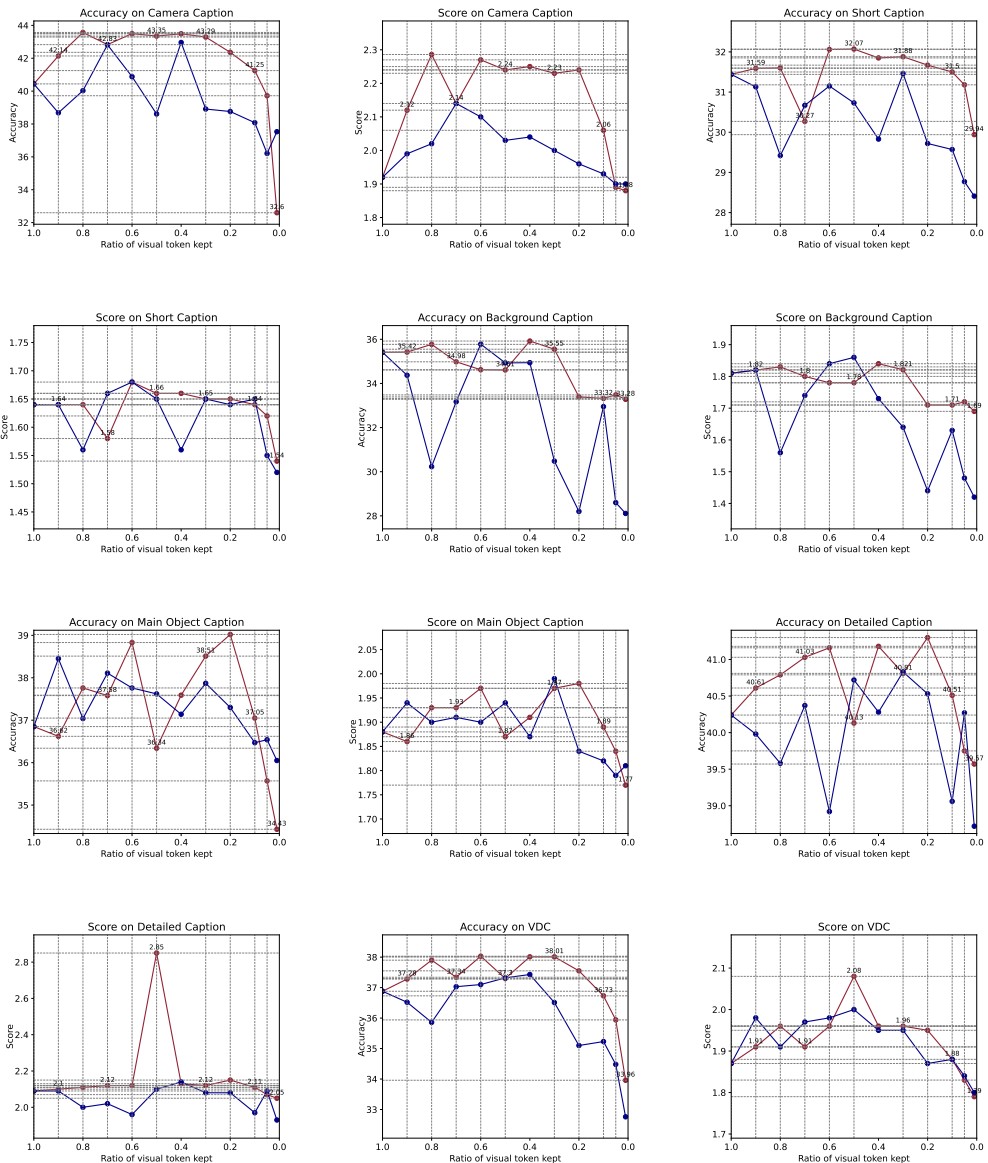

Figure F12: Comparison between performance with and without the inclusion of full first-frame visual tokens during inference on VDC. As the visual token kept ratio varies, the blue curve indicates performance with slowfast inference, while the red curve represents performance without slowfast inference.

Table G9: Training hyper-parameters for AURORACAP.

| Hyper-parameters | Pretraining stage | Vision stage | Language stage |
|---|---|---|---|
| trainable parameters | MLP | ViT + MLP | ViT + MLP + LLM |
| warmup schedule | | linear | |
| warmup start factor | | 1e-5 | |
| warmup ratio | | 0.03 | |
| learning rate schedule | | cosine decay | |
| optimizer | | AdamW (Loshchilov & Hutter, 2017) | |
| optimizer hyper-parameters | | $\beta_1, \beta_2 = (0.9, 0.999)$ | |
| weight decay | | 0.1 | |
| max norm | | 1 | |
| epoch | | 1 | |
| peak learning rate | 2e-4 | 1e-4 | 2e-5 |
| total equivalent batch size | 512 | 6,144 | 768 |
| token keep proportion | 100% | 100% | 10% |

Table H10: Evaluation settings summary for each benchmarks. For all benchmarks we set temperature, top p, number of beams to 0, 0, 1 respectively.

| Benchmark | # Sample | # Tokens | Prompt |
|---|---|---|---|
| Flickr (Plummer et al., 2015) | 31,784 | 64 | Provide a one-sentence caption for the provided image. |
| NoCaps (Agrawal et al., 2019) | 4,500 | 64 | Provide a one-sentence caption for the provided image. |
| COCO-Cap (Lin et al., 2014) | 5,000 | 64 | Provide a one-sentence caption for the provided image. |
| ChartQA (Masry et al., 2022) | 2,500 | 16 | Answer the question with a single word. |
| DocVQA (Mathew et al., 2021) | 5,349 | 32 | Answer the question using a single word or phrase. |
| TextCaps (Sidorov et al., 2020) | 3,166 | 64 | Provide a one-sentence caption for the provided image. |
| GQA (Hudson & Manning, 2019) | 12,578 | 16 | Answer the question using a single word or phrase. |
| POPE (Li et al., 2023h) | 9,000 | 128 | Answer the question using a single word or phrase. |
| MMMU (Yue et al., 2023) | 900 | 16 | Answer with the option letter from the given choices directly. Answer the question using a single word or phrase. |
| VQAv2 (Goyal et al., 2017) | 214,354 | 16 | Answer the question using a single word or phrase. |
| MSR-VTT (Xu et al., 2016) | 1,000 | 64 | Provide a one-sentence caption for the provided video. |
| VATEX (Wang et al., 2019) | 4,478 | 64 | Provide a brief single-sentence caption for the last video below. |
| MSVD-QA (Xu et al., 2017) | 1,161 | 64 | Answer the question using a single word or phrase. |
| ActivityNet-QA (Yu et al., 2019) | 8,000 | 64 | Answer the question using a single word or phrase. |
| MSRVTT-QA (Xu et al., 2017) | 6,513 | 64 | Answer the question using a single word or phrase. |
| iVQA (Yang et al., 2021) | 6,000 | 64 | Answer the question using a single word or phrase. |

# I    LIMITATIONS

Table I11: Comparison AURORACAP with LLM-based SoTA methods on visual question answering benchmarks under zero-shot setting.

| Model | LLM | MMMU (900) Acc | GQA (12,578) Acc | POPE (9,000) F1 | VQAv2 (214,354) Acc |
|---|---|---|---|---|---|
| LLaVA-1.5-7B | Vicuna-1.5-7B | 35.30 | 61.97 | 85.87 | 76.64 |
| LLaVA-1.5-13B | Vicuna-1.5-13B | 34.80 | 63.24 | 85.92 | 78.26 |
| LLaVA-1.6-7B | Vicuna-1.5-7B | 35.10 | 64.23 | 86.40 | 80.06 |
| LLaVA-1.6-13B | Vicuna-1.5-13B | 35.90 | 65.36 | 86.26 | 80.92 |
| AURORACAP-7B | Vicuna-1.5-7B | 36.11 | 59.72 | 83.31 | 75.85 |

Table I12: Limitation in terms of OCR capability compared with LLaVA models. Appendix H shows the introduction and metrics of each benchmark.

| Model | LLM | ChartQA (2,500) Acc | DocVQA (5,349) Acc | TextCaps (3,166) Acc |
|---|---|---|---|---|
| LLaVA-1.5-7B | Vicuna-1.5-7B | 18.24 | 28.08 | 98.15 |
| LLaVA-1.5-13B | Vicuna-1.5-13B | 18.20 | 30.29 | 103.92 |
| LLaVA-1.6-7B | Vicuna-1.5-7B | 54.84 | 74.35 | 71.79 |
| LLaVA-1.6-13B | Vicuna-1.5-13B | 62.20 | 77.45 | 67.39 |
| AURORACAP-7B | Vicuna-1.5-7B | 25.88 | 34.60 | 93.33 |

We evaluate AURORACAP on various visual question answering benchmarks as shown in Table I11. Since the performance of the VQA task heavily depends on the performance of the LLM, we chose the same LLM for a fair comparison. Also, due the the limitation of OCR-related samples in training dataset, AURORACAP does not perform well in OCR at current stage as shown in Table I12.

# J    ADDITIONAL EVALUATION METRICS ANALYSIS

In this section, we make the analysis about more evaluation metrics on VDC other than VDCSCORE. As shown in Figure J13, we found that the correlation between CLIP score and human ratings is low (0.39), which may be due to the fact that, in a lengthy caption, there may be only one or two fine-grained errors that the CLIP text encoder lacks the capability to detect. Meanwhile, we test the impact of different numbers of QA pairs on the results when calculating VDCSCORE. As shown in the Figure J13, more QA pairs produced more robust test results. VDCSCORE shows over 0.86 of pearson correlation with human when using 20 QA pairs.

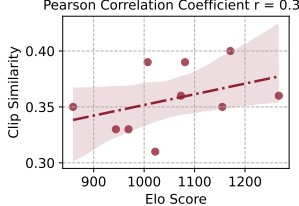 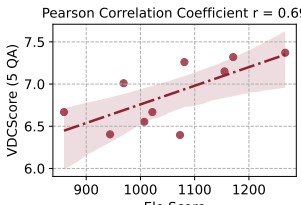 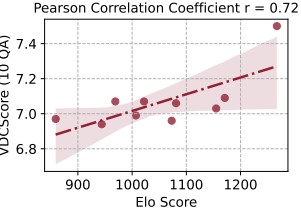

Figure J13: Correlation between the CLIP evaluation scores, different number of QA pairs used in VDCSCORE and human elo ranking on VDC.

## K    VDC GENERATION PROMPT TEMPLATE

Following (Ju et al., 2024), we utilize LLM to generate structured detailed captions. Given an input video, LLM return five detailed captions, including camera caption, short caption, background caption, main object caption and detailed caption for the entire video guided by our designed prompt template. The complete prompt is shown as followings:

| Type | Prompt |
|---|---|
| SYSTEM | You are describing the video. Please provide detailed captions of the video from different aspects. |
| User | Please provide detailed and comprehensive captions for the following content: 1. Short Caption: Summarize the video in one detailed sentence, capturing key actions and the overall mood. 2. Background Caption: Provide a detailed description of the background, including objects, location, weather, time, and any dynamic elements such as movements in the environment. 3. Main Object Caption: Give a thorough description of the main subject's actions, attributes, interactions, and movements throughout the video frames, including changes in posture, expression, or speed. 4. Camera Caption: Describe the camera work in detail, including shot types, angles, movements, transitions, and any special effects used to enhance the video. 5. Reference Caption: Generate a detailed dense caption for the video that is at least 300 words long. The caption should capture all visible actions, environmental details, and the overall emotional atmosphere in depth. Describe in detail the interactions between the main subjects and their environment, including subtle nuances of their movements or expressions. Elaborate on the sounds, textures, and other sensory experiences depicted in the video. Discuss the camera techniques used extensively, including shot types, angles, movements, and transitions. Highlight the mood and tone of the video throughout, creating a rich narrative that connects viewers emotionally to the scene. Include comprehensive descriptions of background elements that add context and depth, such as weather conditions, time of day, and cultural or historical settings. Make sure to provide a vivid portrayal that is engaging, informative, and rich enough for AI to re-generate the video content. 
 No need to provide summary content. Do not describe each frame individually. Avoid using phrases like 'first frame'. The description should be rich enough for AI to re-generate the video. Please generate the response as a Python dictionary string with keys like 'short caption'. DO NOT PROVIDE ANY OTHER OUTPUT TEXT OR EXPLANATION. Only provide the Python dictionary string. 
 These are the frames from the video: [VIDEO KEYFRAME INPUT] |
| GPT-4o | Short Caption: In a sunny outdoor setting, two workers engage in a collaborative effort to prepare a gravel base for a construction project, showcasing teamwork and focus. 
 Camera Caption: The camera captures the scene from a low angle, providing a unique perspective that emphasizes the workers' actions and the surrounding environment. The shots are primarily medium close-ups, focusing on the workers' upper bodies and hands as they interact. The camera occasionally shifts to a wider angle, revealing more of the workspace and background elements. There are smooth transitions between shots, maintaining a steady flow that enhances the viewer's engagement with the unfolding narrative. The use of a slightly fish-eye lens effect adds depth to the scene, making the environment feel more immersive. Background Caption: The scene is set in a residential area with a well-maintained lawn and a brick wall partially constructed in the background. The ground is covered with a white plastic sheet, and scattered gravel is visible. A few wooden planks lie on the ground, and tools are scattered around, including a black water bottle and a power tool. 
 Main Object Caption: The main subjects are two workers, one wearing a gray sweatshirt and gloves, and the other dressed in casual work attire. They are actively engaged in conversation, with the first worker gesturing with his hands to emphasize points while explaining the task at hand. His posture is relaxed yet focused, indicating a collaborative atmosphere. The second worker listens attentively, occasionally nodding and responding with gestures, showcasing a dynamic interaction. Both workers exhibit a sense of camaraderie and professionalism as they prepare for the next steps in their project. Reference Caption: The video unfolds . . . (300 words) . . . through them. |
| User | The video has been describe from the following aspects:1. short caption: short caption, 2. background caption: background caption, 3. main object caption: main object caption, 4. camera caption: camera caption, 5.Reference Caption: reference caption Please generate a detailed dense caption for the video that is pretty long. You should expand the reference caption upon the information provided in the short caption, background caption, main object caption, and camera caption. Ensure that the detailed caption does not introduce any new entities or relationships that were not mentioned in the previous captions. Make sure to provide a vivid portrayal that is engaging, informative, and rich enough for AI to re-generate the video content. "Avoid using phrases like 'first frame', 'short caption', 'background caption', 'main object caption', and 'camera caption'. The description should be rich enough for AI to re-generate the video. |
| GPT-4o | Detailed Caption: The video opens with . . . (500 words) . . . literature that surrounds them. |
| User | [Manually check and refine] |

## L QUESTION-ANSWER PAIRS GENERATION PROMPT TEMPLATE OF VDCSCORE

To decompose the ground-truth structured detailed captions in VDC, we utilize `GPT-4o` as the LLM assistant to generate numerous short question-answer pairs for subsequent evaluation. The full prompt and example cases are presented as followings:

| Type | Prompt |
|---|---|
| SYSTEM | You are an intelligent chatbot designed for generating 20 question-answer pairs given a detailed description of a video or image. You are describing the video.
Here's how you can accomplish the task: INSTRUCTIONS: - Cover the main objects and actions in the video or image.
- The questions should be open-ended and start with 'What', 'Who', 'Where', 'When', 'Why', 'How', etc.
- The answer should be a short sentence or phrase.
- Generate 20 question-answer pairs. |
| User | Please generate 20 question-answer pairs given a detailed description of a video or image: detailed description: The camera work is dynamic, featuring a mix of close-up shots and wider angles that provide context to the environment. The shots are slightly shaky, adding a sense of realism and immersion to the viewer. Transitions between shots are smooth, with the camera often panning or tilting to follow the subject's movements. The use of low light creates a moody atmosphere, with shadows enhancing the visual interest of the scene ...
Please generate the response in the form of a Python list of tuple with the question and the corresponding answer. DO NOT PROVIDE ANY OTHER OUTPUT TEXT OR EXPLANATION. Only provide the Python list of tuple. For example, your response should look like this: [(the question 1, the answer 1), (the question 2, the answer 2), ... ]. |
| GPT-4o | ["question": "What is the style of camera work in the video?", "answer": "dynamic", "question": "What is the effect of shaky shots in the video?", "answer": "realism and immersion", "question": "What type of transitions are used in the video?", "answer": "smooth", "question": "Why is the moody atmosphere created in the video?", "answer": "through low light", "question": "What type of camera movement is achieved through panning and tilting?", "answer": "following the subject's movements", "question": "What is the effect of the camera movement on the viewer?", "answer": "sense of realism and immersion", ... ] |
| User | [Manually check and refine] |

## M MORE STATISTICS INFORMATION OF VDC

Figure M14 indicates the word distribution of the structured captions in VDC. Table M13 illustrates the distribution of the question categories we provided for VDCSCORE calculation.

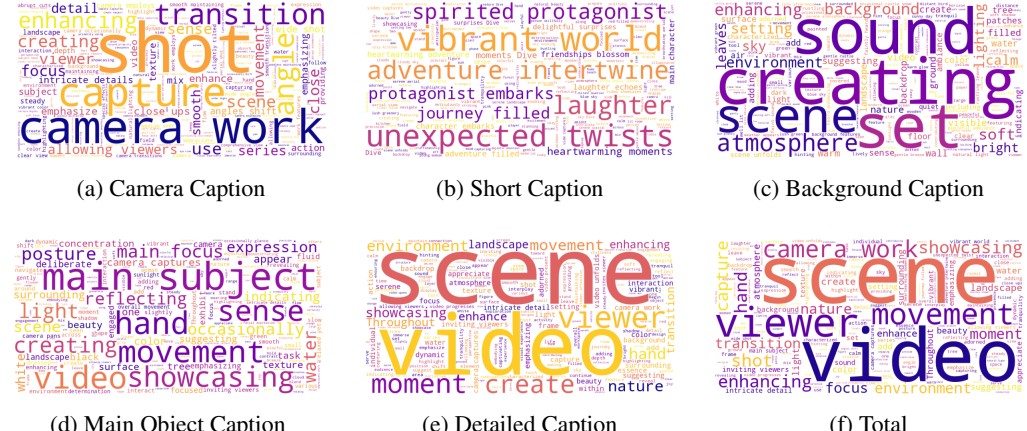

(a) Camera Caption     (b) Short Caption     (c) Background Caption

(d) Main Object Caption     (e) Detailed Caption     (f) Total

Figure M14: Word cloud of different structured captions in VDC, showing the diversity.

Table M13: Distribution of the question categories in VDCSCORE.

| Type | Background | Camera | Short | Main object | Detailed |
|---|---|---|---|---|---|
| Environment & Scene | 3,464 | 2,543 | 4,401 | 3,097 | 3,549 |
| Character & Object | 6,272 | 3,779 | 4,834 | 6,035 | 6,440 |
| Intent & Outcomes | 4,585 | 5,105 | 5,067 | 4,474 | 4,640 |
| Action | 2,861 | 1,198 | 1,857 | 3,192 | 1,932 |
| Attributes & Relationship | 944 | 334 | 1,388 | 1,478 | 1,280 |

## N    MORE STATISTICS INFORMATION OF VDCSCORE

We generate a total of 96,902 question-answer pairs for VDC, with an average of 18.87 pairs per detailed caption. As depicted in Figure N15, each section of the structured captions includes a similar number of question-answer pairs. Additionally, Figure N16 presents the distribution of question types generated for VDC. To enhance the evaluation of detailed captions, we configure all questions as open-ended. `Environment & Scene` encompasses inquiries about location, environment, atmosphere, scene, and time. `Character & Object` focuses on entities within the caption, such as people, animals, plants, and objects, while `Attribute & Relation` examines their attributes and interrelations. `Intent & Outcomes` addresses deeper interpretative questions regarding methods, purposes, reasons, and outcomes. We further analyze the distribution of generated question types within structured captions. Main object captions predominantly feature `Character & Object` questions, whereas `Environment & Scene` questions are more prominent in background captions, and `Camera` questions constitute a larger proportion in camera.

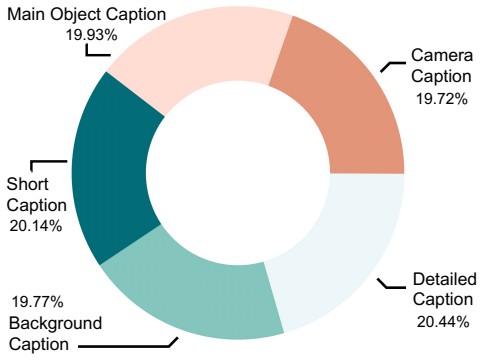

Figure N15: Question-answer pairs proportaion in structured captions.

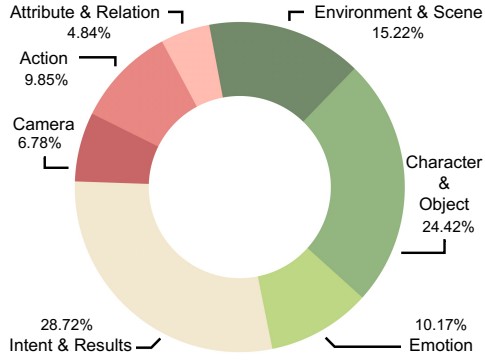

Figure N16: Distribution of question type in VDCSCORE.

## O    CALCULATION OF ELO RANKING

In this section, we present the methodology for evaluating and ranking AURORACAP and various models using the Elo rating system as shown in Figure O17. The parameters used in the simulation are summarized in Table O14.

1. For all models participating in the ELO ranking, we collecte the output captions for each video.
2. As shown in Figure O17, we develop a frontend tool to randomly select captions generated by two different models for the same video. Without revealing the model IDs, human evaluators then chose the caption they found better.
3. We record comparison results and calculated the ELO values based on the parameters in Table O14.
4. In Figure 5, we calculate the Pearson correlation between different metrics and human ELO values, ultimately showing that our proposed VDC metric is the closest to human evaluations.

Table O14: Elo parameter setting.

| Parameter | Number |
|---|---|
| initial Elo mean | 1,000 |
| Elo standard deviation | 300 |
| base of logarithm | 10 |
| scaling factor | 400 |
| K-factor | 32 |
| minimum Elo rating | 700 |
| number of simulated matches | 2,778 |

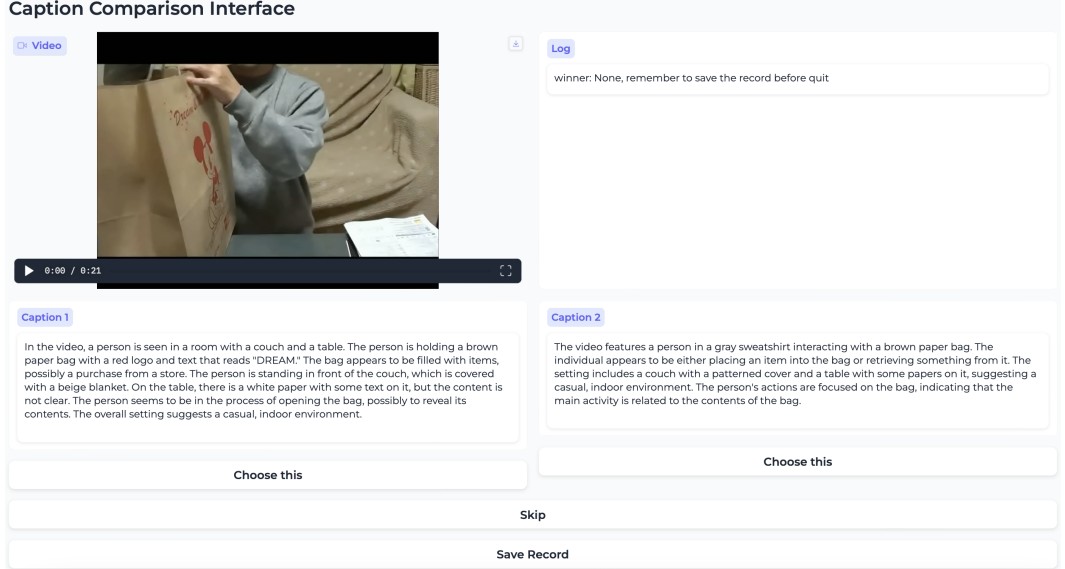

Figure O17: GUI screen for Elo ranking.

# P CASE STUDY

We perform an extensive case study of AURORACAP on a variety of videos for video detailed captioning. As shown as followings, AURORACAP is capable of providing excellent detailed captions regarding the camera motion, background and main object with less hallucination.

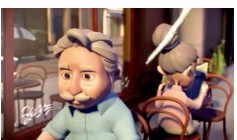 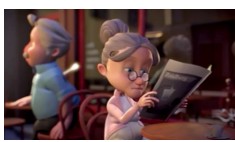 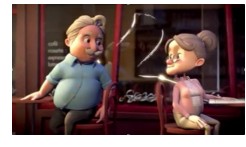 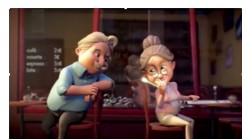

Figure P18: Video example of MSR-VTT (Xu et al., 2016) benchmark.

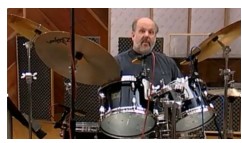 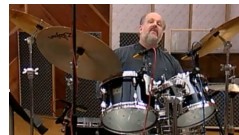 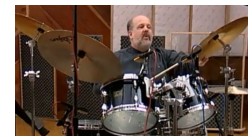 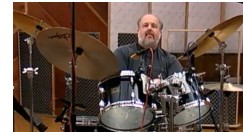

Figure P19: Video example of VATEX (Wang et al., 2019) benchmark.

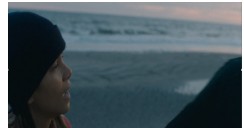 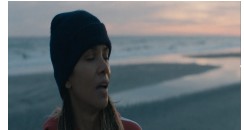 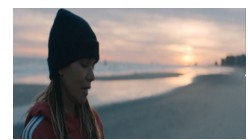 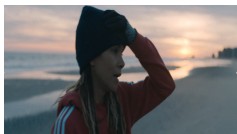

Figure P20: Video example.

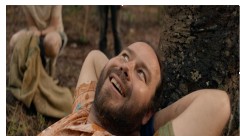 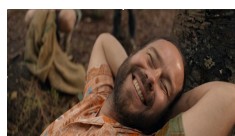 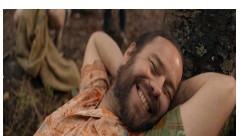 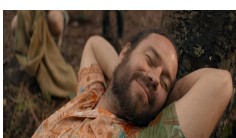

Figure P21: Video example.

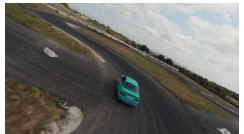 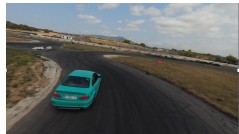 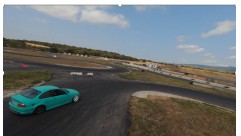 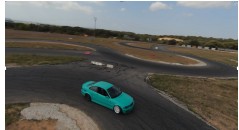

Figure P22: Video example.

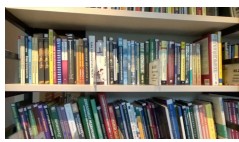 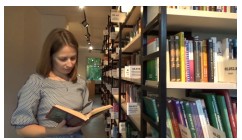 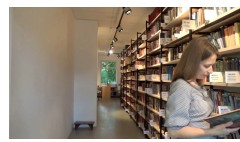 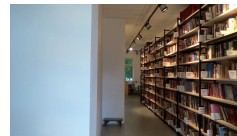

Figure P23: Video example.

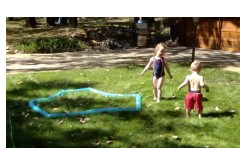 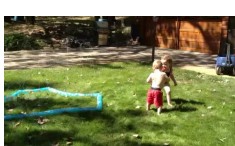 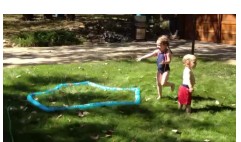 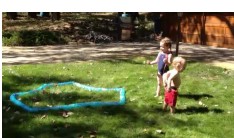

Figure P24: Video example.

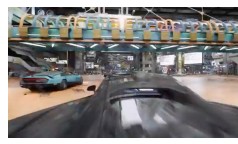 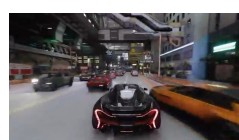 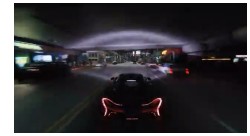 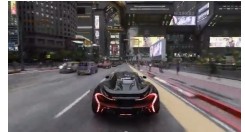

Figure P25: Video example.

**Example  Caption**

Figure P18  [short caption] A man with a mustache and a woman with a ponytail are sitting at a table.

Figure P19  [short caption] A man plays the drums in a recording studio.

Figure P20  [detailed caption] The video depicts a serene beach scene where a young woman stands on the sandy shore, gazing out towards the ocean. She is wearing a black beanie and a pink jacket, adding a pop of color to the otherwise muted scene. The beach, a vast expanse of sand, stretches out

in front of her, meeting the ocean at the horizon. The ocean, a vast body of water, is visible in the background. The beach is bathed in a soft, diffused light, creating a dreamy atmosphere. The girl's gaze is directed towards the horizon, suggesting a sense of wonder or contemplation. The image is slightly blurred, adding a dreamy quality to the scene. The woman's position on the beach, coupled with the gentle waves of the ocean, suggests a moment of contemplation or admiration. The relative positions of the objects suggest a peaceful day at the beach, with the girl possibly enjoying the serene view of the ocean. The colors are mostly muted, with the girl's pink jacket standing out against the sandy beach and the blue ocean. The blurred background and the out-of-focus elements, such as the ocean and the sky, contribute to the sense of tranquility and focus on the woman. There is no text present in the video, and the colors are muted, with the exception of the pink jacket, which stands out against the more subdued tones of the surroundings.

Figure P21 [detailed caption] The video features a man with a beard and long hair, lying on the ground with his head resting on a tree trunk. He is wearing a colorful shirt with a mix of orange and green patterns. The man's face is partially obscured by the tree trunk, but his eyes are visible, looking upwards. He appears to be smiling or laughing, with his mouth open. The man's arms are crossed over his chest, suggesting a relaxed or playful posture. The background is blurred, but it appears to be a natural outdoor setting, possibly a forest or park. In the background, another person is partially visible, their presence subtly hinted at by a hand reaching out from the left side of the frame. The colors in the video are warm, with the man's shirt standing out against the muted background. The man's hair and beard are dark, and he has a light complexion. The overall mood of the video is lighthearted and carefree. There are no visible texts or other objects in the video, and the relative positions of the objects remain constant with the man in the foreground and the second person in the background.

Figure P22 [background caption] The background features a race track with visible tire marks and barriers, surrounded by grassy areas and a few scattered trees. The track is set in a rural or semi-rural location, with hills in the distance and a cloudy sky overhead, suggesting overcast weather conditions. The track itself appears well-maintained with a smooth surface, designed for high-speed racing. The weather, indicated by the cloudy sky, contributes to the overall ambiance of the scene, enhancing the sense of a cool, possibly early morning or late afternoon setting.

Figure P23 [camera caption] The camera maintains a steady, frontal view throughout the video, capturing the woman's expressions and the intricate details of the bookshelves. The camera starts at the right edge of the bookshelf, moving across the shelf, and ends at the left edge of the bookshelf. The framing is consistent, focusing on the woman and the bookshelves, with the camera positioned at a medium distance to the subject. This movement provides a comprehensive view of the bookshelf, showcasing the variety of books and their arrangement on the shelves. The camera occasionally pans to reveal the depth of the library, showcasing the rows of books and the inviting atmosphere. The use of natural light enhances the visual appeal, creating a warm and inviting tone throughout the video.

Figure P24 [camera caption] The view shot remains relatively static, focusing on the children playing in the backyard. The camera angle is at eye level, capturing the scene from a distance that allows both children to be visible. There is minimal camera movement, maintaining a steady focus on the children and their activities. The sequence of video frames suggests a continuous moment of play without significant changes in shooting angles or camera movement.

Figure P25 [main object caption] The main subject in the video, a black car, is seen driving down a street that appears to be in a state of disarray. The car moves steadily forward, navigating around obstacles such as a blue car parked on the side of the road. The car's movement is smooth and continuous, suggesting it is either in motion or has just come to a stop. The environment around the car is chaotic, with debris scattered across the road and signs of destruction, indicating a recent event or disaster. The car's position remains central in the frame, with the camera angle focused on it from a slightly elevated perspective, possibly from a vehicle or a structure above.

# Q   PREDICTED ANSWER EXTRACTION PROMPT TEMPLATE

Given the question-answer pairs based on the ground truth caption, we utilize `GPT-4o` to extract predicted answers based on the generated caption by our designed prompt template. The complete prompt is shown as followings:

**Type  Prompt**

SYSTEM  You are an intelligent chatbot designed for providing accurate answers to questions related to the content based on a detailed description of a video or image.
Here's how you can accomplish the task:"
——

##INSTRUCTIONS:
- Read the detailed description carefully.
- Answer the question only based on the detailed description.
- The answer should be a short sentence or phrase.

User  Please provide accurate answers to questions related to the content based on a detailed description of a video or image:

detailed description: The setting is an urban street with a distinctive red and white painted bike lane, which is part of a larger roadway. The lane is marked with white lines and the word "WALK" is painted in large, bold letters, indicating a pedestrian crossing area. The pavement is a mix of red brick and grey tiles, with a curb separating the bike lane from the adjacent road. The weather appears to be clear and sunny, casting sharp shadows on the ground, suggesting it is daytime. There are no other vehicles or pedestrians in sight, and the road is empty, which could imply a quiet time of day or a less busy area. The presence of a yellow circular object on the handlebar of the bicycle is notable, possibly a bell or a light.

question: What objects are the cyclist's hands interacting with?

DO NOT PROVIDE ANY OTHER OUTPUT TEXT OR EXPLANATION. Only provide short but accurate answer.

GPT-4o  The cyclist's hands are interacting with the bicycle handlebars.

## R  CORRECTNESS EVALUATION PROMPT TEMPLATE

Following (Maaz et al., 2023), we evaluate the correctness and score of the predicted answers with the assistant of GPT-4o. Given the question, correct answer, and predicted answer from the generated caption, GPT-4o should return the *True* or *False* judgement and relative score (0 to 5). The complete prompt is shown as followings:

**Type  Prompt**

SYSTEM  You are an intelligent chatbot designed for evaluating the correctness of generative outputs for question-answer pairs.
Your task is to compare the predicted answer with the correct answer and determine if they match meaningfully. Here's how you can accomplish the task:
——

##INSTRUCTIONS:
- Focus on the meaningful match between the predicted answer and the correct answer.
- Consider synonyms or paraphrases as valid matches.
- Evaluate the correctness of the prediction compared to the answer.

User  Please evaluate the following video-based question-answer pair:
Question: What type of surface can be seen covering the floor?
Correct Answer: carpet
Predicted Answer: Carpeted surface.
Provide your evaluation only as a yes/no and score where the score is an integer value between 0 and 5, with 5 indicating the highest meaningful match.
Please generate the response in the form of a Python dictionary string with keys 'pred' and 'score', where value of 'pred' is a string of 'yes' or 'no' and value of 'score' is in INTEGER, not STRING.
DO NOT PROVIDE ANY OTHER OUTPUT TEXT OR EXPLANATION. Only provide the Python dictionary string.
For example, your response should look like this: {'pred': 'yes', 'score': 4.8}.

GPT-4o  {'pred': 'yes', 'score': 5}

