# OpenReview forum: "AuroraCap: Efficient, Performant Video Detailed Captioning and a New Benchmark"
_ICLR.cc/2025/Conference — ICLR 2025 Poster_

### Official Review · Reviewer_ZSxM · 2024-10-30

**Soundness:** 3
**Presentation:** 3
**Contribution:** 3
**Rating:** 6
**Confidence:** 4

**Summary:**

The paper introduces AuroraCap, a video captioning model based on LMM, along with VDC, a detailed captioning benchmark, and VDCScore, an evaluation metric designed for long captions. AuroraCap handles video sequences with lower visual token numbers by the token merge strategy, outperforming state-of-the-art models, even including GPT-4V and Gemini-1.5 Pro.  VDC addresses the limitations of existing video captioning benchmarks by providing over one thousand videos with detailed, structured captions. since existing metrics are insufficient to evaluate detailed captioning performance, they develop a new LLM-assisted metric VDC-SCORE, that breaks down long captions into short question-answer pairs, ensuring better alignment with human judgments.

**Strengths:**

1. Despite adopting a simple architecture design consistent with LLaVA, AuroraCap achieves remarkable results on captioning tasks with its three-stage training recipe.
2. AuroraCap combines the Token Merging strategy, significantly reducing the number of visual tokens input and thereby improving inference efficiency.
3. This paper also presents VDC, a detailed video description benchmark, which includes over one thousand structured descriptions, providing detailed annotations for videos from multiple perspectives.
4. To accurately evaluate the model’s detailed captioning capabilities, the paper introduces VDCScore. This metric uses a divide-and-conquer strategy to break long captions into short QA pairs and employs LLM assistance for scoring. This approach avoids the challenges traditional metrics face with long captions and mitigates potential hallucination issues associated with using LLM evaluations.
5. Compared to existing models, AuroraCap achieves excellent performance in both image captioning and video captioning tasks.
6. The paper also explores the impact of the visual token kept ratio on model performance, demonstrating that with the token merging strategy, fewer visual tokens can still maintain high performance.

**Weaknesses:**

1. **Spatial Token Merging:** AuroraCap only performs spatial token merging within frames and does not implement temporal token merging between frames, which could further enhance inference efficiency.
2. **Static Token Merging:** It seems that for different visual inputs, the number of visual tokens remains consistent by setting a visual token kept ratio. However, this ‘static’ token merging strategy is unreasonable for tokens with varying levels of informativeness.
3. **Unclear Training Strategy:** It is not clear whether the token merging strategy was used during training. Additionally, the impact of using different token merge ratios during training on inference performance is not clear.
4. **Performance in Video Question Answering:** Although AuroraCap demonstrates leading performance in captioning tasks, its advantages in video question answering tasks are not apparent, which is perplexing.

**Questions:**

1. **Training Strategy Clarity:** Was the token merging strategy used during AuroraCap’s training phase? How does varying the token merge ratio during training impact inference performance?
2. **Performance in Video Question Answering:** What factors limit AuroraCap’s performance in video question answering compared to captioning tasks?
3. **Token Merge Curve on VDC** As shown in Figure 6, why does retaining all visual tokens not lead to the highest VDCScore?

I will consider raising my score if the authors can further address my concerns.

---

> ### Author Response · Authors · 2024-11-16
> **Response to Weaknesses 1-2**
>
> Thank you for your constructive comments. We have carefully considered your suggestions and would like to provide the following
> clarifications. We have also updated the manuscript, with the modified contents highlighted in orange for your review.
>
> ---
>
> **Q1: Spatial Token Merging: AuroraCap only performs spatial token merging within frames and does not implement temporal token merging between frames, which could further enhance inference efficiency.**
>
> A1: In our current implementation, we focus on spatial token merging to simplify the model and demonstrate the effectiveness of token reduction within individual frames. We agree that incorporating temporal token merging could further enhance inference efficiency by reducing redundancy across frames. Existing methods have made preliminary explorations in this direction, and we plan to explore temporal token merging to optimize AuroraCap’s performance without compromising the temporal dynamics essential for video understanding.
>
> In fact, for current video inputs, when we sample at 1 FPS, the frames are already sufficiently sparse; frame-to-frame similarity is quite low except in static scenes. On the other hand, if we use keyframes as input, the frame similarity will be even lower. Under these conditions, we believe that strong temporal merging should be avoided, as it is likely to impair the model's core capabilities, especially since the vision encoder we use is a CLIP trained on single frames. As suggested by LongVU [1], we believe a more effective approach is to merge only spatially and apply a query-based selection method in the temporal dimension.
>
> **Table. Performance of temporal merging on different merge layers on Egoschema[2]. We merge each frame with the visual token kept ratio of 0.4.**
> | Merge Layer |  Egoschema Acc.  |
> |------------|------------------|
> | 0          | 29.6%            |
> | 7          | 33.4%            |
> | 15         | 38.2%            |
> | 23         | 39.2%            |
> | 30         | 30.2%            |
> | Without Merge | 45.0%         |
>
> We find that no metter where we conduct temporal merging, the performance is always lower than spatial merging only.
>
> ---
>
> **Q2: Static Token Merging: It seems that for different visual inputs, the number of visual tokens remains consistent by setting a visual token kept ratio. However, this ‘static’ token merging strategy is unreasonable for tokens with varying levels of informativeness.**
>
> A2: Thank you for pointing out this limitation. Our static token merging strategy is intend to simply establish a baseline for performance improvement through token reduction. We acknowledge that different inputs contain varying amounts of information, and a dynamic token merging strategy could better preserve critical visual details as shown in Appendix E. We observe that a simple image with few tokens (6 tokens) conveys rich information (row 1 of Figure C4 (Figure E6 in the current version)), while a complex image with more tokens (154 tokens) fails to do so (row 2 of Figure C4 (Figure E6 in the current version)), highlighting the necessity of Static Token Merging.
>
> We agree that dynamically adjusting the token merge ratio is a feasible approach—for example, applying a more moderate merging ratio for OCR and VQA tasks. Based on experience, we recommend using a token_kept_ratio in the range of 0.2 to 0.4 for a better performance-cost trade-off in captioning tasks, above 0.5 for visual question answering tasks, and above 0.8 for OCR-related tasks. However, enabling the model to automatically select the optimal ratio is challenging and may require further training or the addition of new modules. To keep the baseline simple, we consider this a direction for future work and encourage the community to explore it. In future work, we intend to develop adaptive token merging mechanisms that assess token informativeness, possibly leveraging attention weights or other relevance metrics to retain the most significant tokens dynamically.

---

> ### Author Response · Authors · 2024-11-16
> **Response to Weaknesses 3-4 and Qustions**
>
> **Q3: Unclear Training Strategy: It is not clear whether the token merging strategy was used during training. Additionally, the impact of using different token merge ratios during training on inference performance is not clear.**
>
> A3: We apologize for the lack of clarity regarding our training strategy. The token merging strategy is indeed applied during both the language stage of training and the inference phases. We employ token merging in the language stage with the visual token kept ratio set to 0.1 to achieve efficient training. We also experiment by varying the training visual token kept ratio to 1 to explore its impact on inference. As shown in Figure D6 (Figure F8 in the current version) in the Appendix, Setting A corresponds to a visual token kept ratio of 1 during the language stage, while Setting C corresponds to a visual token kept ratio of 0.1. We observe that across six different tasks, different training visual token kept ratios leeds to similar performance on downstream tasks. However, Setting A resulted in a much longer training time, indicating that using a visual token kept ratio of 1 during the language stage is a suboptimal choice. We will include detailed explanations and additional experimental results in the revised paper to clarify this aspect.
>
> We apologize for the lack of clarity regarding our training strategy. The token merging strategy is indeed applied during both the language stage of training and the inference phases. We employ token merging in the language stage with the visual token kept ratio set to 0.1 to achieve efficient training. We also experiment by varying the training visual token kept ratio to 1 to explore its impact on inference. As shown in Figure D6 (Figure F8 in the current version) in the Appendix, Setting A corresponds to a visual token kept ratio of 1 during the language stage, while Setting C corresponds to a visual token kept ratio of 0.1. We observe that across six different tasks, different training visual token kept ratios leeds to similar performance on downstream tasks. However, Setting A resulted in a much longer training time, indicating that using a visual token kept ratio of 1 during the language stage is a suboptimal choice. We will include detailed explanations and additional experimental results in the revised paper to clarify this aspect.
>
> ---
> **Q4: Performance in Video Question Answering: Although AuroraCap demonstrates leading performance in captioning tasks, its advantages in video question answering tasks are not apparent, which is perplexing.**
>
> A4: We appreciate your observation. As AuroraCap is primarily designed as a captioning model, the training data for the video version of AuroraCap mainly focuses on video captions, primarily sourced from MiraData. AuroraCap is primarily designed and optimized for detailed video captioning, and our training data and loss functions were tailored accordingly. The nuances of video question answering require different reasoning capabilities of LLMs and often demand understanding specific details prompted by the questions.  Therefore, AuroraCap doesn’t perform ideally on video question answering tasks. In our future work, we plan to incorporate more video question answering data, along with other academic datasets, to comprehensively improve the model’s performance across various tasks.
>
> ---
> **Q5: Token Merge Curve on VDC As shown in Figure 6, why does retaining all visual tokens not lead to the highest VDCScore?**
>
> A5: We appreciate your insightful observation. Unlike in image scenarios, including all tokens of a video may introduce noise and redundant information, making it harder for the model to focus on the most salient features of the video. Furthermore, since AuroraCap is trained with a visual token kept ratio of 0.1 during the language stage, the model learns to describe more with fewer tokens. With more tokens during inference, the model might overlook important aspects or focus on redundant details, which doesn’t generalize well to the evaluation metrics. Our findings suggest that an optimal token kept ratio exists where the model achieves a balance between retaining essential information and discarding redundant data, leading to better overall performance, especially for video scenarios. We will expand our analysis in the revised paper to provide a deeper understanding of this phenomenon.
>
> ---
> **Reference**
>
> [1] Shen, Xiaoqian, et al. "LongVU: Spatiotemporal Adaptive Compression for Long Video-Language Understanding." arXiv preprint arXiv:2410.17434 (2024).
>
> [2] EgoSchema: A Diagnostic Benchmark for Very Long-form Video Language Understanding. https://egoschema.github.io

---

### Official Review · Reviewer_gKDb · 2024-10-30

**Soundness:** 3
**Presentation:** 3
**Contribution:** 3
**Rating:** 6
**Confidence:** 3

**Summary:**

This paper proposes AURORACAP, a video captioner based on a large multimodal model that utilizes a token merge strategy to support lengthy video sequences. The authors also introduce VDC, a video detailed captioning benchmark featuring over one thousand carefully annotated structured captions. Finally, the authors validate the effectiveness of their model through a divide-and-conquer approach across multiple benchmarks, including the newly introduced VDC and various metrics, including human Elo ranking.

**Strengths:**

1. This work introduces the application of a token merging strategy for video understanding, demonstrating through extensive experiments that it can utilize only 10% to 20% of the visual tokens compared to the original tokens generated by ViT, with only a marginal drop in performance across various benchmarks.
2. The proposed VDC benchmark addresses dimensions that are lacking or insufficient in existing benchmarks, such as rich world knowledge, object attributes, camera movements, and, crucially, detailed and precise temporal descriptions of events.
3. This paper is well-structured and includes intuitive figures that effectively illustrate the work.

**Weaknesses:**

1. There is a lack of novelty regarding the token merging strategy, as similar attempts have already been made in Chat-UniVi[1], which validated the effectiveness of this approach on video understanding benchmarks. Furthermore, the divide-and-conquer validation of the model essentially constructs Video QA based on video detail captioning, a method that has also been explored in [2].
2. It would be beneficial to evaluate the token merging method's stability on longer video sequence benchmarks, such as Egoschema[3] and VideoMME[4].

[1] Chat-UniVi: Unified Visual Representation Empowers Large Language Models with Image and Video Understanding. https://arxiv.org/abs/2311.08046
[2] Direct Preference Optimization of Video Large Multimodal Models from Language Model Reward. https://arxiv.org/abs/2404.01258
[3] EgoSchema: A Diagnostic Benchmark for Very Long-form Video Language Understanding. https://egoschema.github.io
[4] Video-MME: The First-Ever Comprehensive Evaluation Benchmark of Multi-modal LLMs in Video Analysis. https://arxiv.org/pdf/2405.21075

**Questions:**

1. Why do the authors train their model on general tasks while naming it a captioner? It seems more reasonable to refer to it as a general model while emphasizing its captioning capabilities.
2. Why does GPT-4V have the lowest CIDEr score on the Flickr benchmark? I believe the authors need to verify whether a fair comparison was conducted, including aspects such as the prompts used during evaluation. I do not consider this to be a reasonable outcome.

---

> ### Author Response · Authors · 2024-11-16
> **Response to Weaknesses 1-2**
>
> Thank you for your constructive comments. We have carefully considered your suggestions and would like to provide the following
> clarifications. We have also updated the manuscript, with the modified contents highlighted in orange for your review.
>
> ---
> **Q1: There is a lack of novelty regarding the token merging strategy, as similar attempts have already been made in Chat-UniVi[1], which validated the effectiveness of this approach on video understanding benchmarks. Furthermore, the divide-and-conquer validation of the model essentially constructs Video QA based on video detail captioning, a method that has also been explored in [2].**
>
> A1: Thank you for the discussion on related work. The focus of this paper is to introduce a new video detailed captioning task along with a solid baseline; thus, we did not pursue novelty in the baseline extensively. Instead, we aimed to keep it straightforward and robust.
>
> Regarding the token merging, we referenced Chat-UniVi [1] in the related work. Compared to this, the main difference in our model lies in more extensive experiments, as illustrated in Figures 4, 5, D7, D8, D9, and D10 across a wide range of tasks, including image captioning and QA, video captioning, QA, and detailed captioning, and analyzed the impact of varying the token kept ratio from 0 to 1 on the results. Additionally, in Figures D5 and D6, we demonstrated that token merging is effective in both training and testing stages, enhancing efficiency without compromising performance. In terms of implementation details, Chat-UniVi uses KNN to identify redundant tokens, whereas we found that a simple bipartite graph construction can achieve the same effect, with greater efficiency and easier control. Besides, Chat-UniVi conduct temporal merging but we don't. In fact, for current video inputs, when we sample at 1 FPS, the frames are already sufficiently sparse; frame-to-frame similarity is quite low except in static scenes. We believe that strong temporal merging should be avoided, as it is likely to hurt the model's performance, especially since CLIP vision encoder is trained on single frame. We believe a more effective approach is to merge only spatially and apply a query-based selection method in the temporal.
>
> We referenced LLAVA-HOUND-DPO [2] and its training data in the related work section, as we believe this is also an effective strategy. The difference lies in the motivation: LLAVA-HOUND-DPO uses this strategy to build a preference dataset, ensuring that instructional data remains factually consistent with detailed captions. Our goal, however, is to evaluate the accuracy of generated captions through QA pairs generated from ground truth. Therefore, in our approach, we have two captions that we need to compare in terms of similarity, rather than generating multiple QA pairs from a single caption.
>
> In the updated manuscript, we provide a more detailed discussion of the differences between our work and these references on page 24.
>
> ---
> **Q2: It would be beneficial to evaluate the token merging method's stability on longer video sequence benchmarks, such as Egoschema[3] and VideoMME[4].**
>
> A2: Thank you for your suggestion regarding the long video understanding tasks. We take Egoschema[3] as the representative benchmark, and compare with existing sota models. As shown in the following table, AuroraCap achieves comparable performance even without training on long videos and special design. We also plot the visual token kept ratio curve on EgoSchema[3], demonstrating the stability and effectiveness of our token merging method for longer video sequences. We include these evaluations in the appendix in revised manuscript on page 23-24 as your recommendation.
>
> **Comparison of different models on Egoschema[3]. Without training on long videos and special design, AuroraCap achieves comparable performance.**
> |Model|Long Video|Acc|
> |-|-|-|
> |Random Choice|-|20.0%|
> |FrozenBiLM|✗|26.9%|
> |mPLUG-Owl|✗|31.1%|
> |TimeChat|✓|33.0%|
> |Video-LLAVA|✗|38.4%|
> |LLAMA-VID|✗|38.5%|
> |LLAVA-NeXT-Video|✗|43.9%|
> |**AuroraCap (ours)**|✗|46.0%|
> |VideoLLAMA2|✗|51.7%|
> |MovieChat|✓|53.5%|
> |Human Performance|-|76.2%|
>
> **Comparison of different models on MovieChat-1K[5]. AuroraCap achieves the best performance on Breakpoint Acc, and comparable performance on Global Acc.**
> |Model|Long Video|Breakpoint Acc.|Global Acc.|
> |-|-|-|-|
> |Video-LLaMA|✗|39.1|51.7|
> |VideoChat|✗|46.1|57.8|
> |TimeChat|✓|46.1|73.8|
> |VideoChatGPT|✗|48.0|47.6|
> |MovieChat|✓|48.3|62.3|
> |MovieChat+|✓|49.6|71.2|
> |**AuroraCap (ours)**|✗|52.6|59.7|
>
> **Effect of visual token kept ratio on Egoschema[3]. We observe that the performance achieves the best when the visual token kept ratio is between 0.6 and 0.7.**
> |Visual Token Kept Ratio|0.01|0.05|0.1|0.2|0.3|0.4|0.5|0.6|0.7|0.8|0.9|1.0|
> |-|-|-|-|-|-|-|-|-|-|-|-|-|
> |Egoschema|38.6|41.8|43.4|43.2|44.8|45.0|45.6|**46.0**|**46.0**|45.0|41.2|40.2|

---

> ### Author Response · Authors · 2024-11-16
> **Response to Questions 1-2**
>
> **Q3: Why do the authors train their model on general tasks while naming it a captioner? It seems more reasonable to refer to it as a general model while emphasizing its captioning capabilities.**
>
> A3: Thank you for your question regarding the model's capabilities. Yes, although AuroraCap was primarily designed for captioning, we did incorporate data from various other tasks during training, such as VQA, classification, and more. During the vision stage of training, we even used data from detection, depth estimation, and segmentation tasks to train the ViT component. While these tasks are not specifically for captioning, they can positively contribute to the training process. Finally, we agree with your suggestion to "refer to it as a general model while emphasizing its captioning capabilities," as it’s a more precise description, and we include this in the revised manuscript Line 112.
>
> ---
> **Q4: Why does GPT-4V have the lowest CIDEr score on the Flickr benchmark? I believe the authors need to verify whether a fair comparison was conducted, including aspects such as the prompts used during evaluation. I do not consider this to be a reasonable outcome.**
>
> A4: Thank you for your concern regarding this issue. We update the manuscript Line 164 with the following clarifications. For GPT-4V, our data source is Table 7, "Model Performances on Image-to-Text Capabilities," from reference [6]. We have faithfully reported their test results, which, according to their description, are based on 8-shot API testing. It’s worth mentioning that this is not a critical dataset, as our approach mainly focuses on video tasks and detailed captioning, while these benchmarks consist of single-sentence image captions. This difference might also explain why GPT-4V performs less effectively on this task.
>
> ---
> **Reference**
>
> [1] Chat-UniVi: Unified Visual Representation Empowers Large Language Models with Image and Video Understanding. https://arxiv.org/abs/2311.08046
>
> [2] Direct Preference Optimization of Video Large Multimodal Models from Language Model Reward. https://arxiv.org/abs/2404.01258
>
> [3] EgoSchema: A Diagnostic Benchmark for Very Long-form Video Language Understanding. https://egoschema.github.io
>
> [4] Video-MME: The First-Ever Comprehensive Evaluation Benchmark of Multi-modal LLMs in Video Analysis. https://arxiv.org/pdf/2405.21075
>
> [5] Moviechat: From dense token to sparse memory for long video understanding. https://github.com/rese1f/MovieChat
>
> [6] Team, Chameleon. "Chameleon: Mixed-modal early-fusion foundation models." arXiv preprint arXiv:2405.09818 (2024). link: https://arxiv.org/pdf/2405.09818#page=17

---

> ### Comment · Reviewer_gKDb · 2024-11-22
>
> My concerns have been addressed, so I will increase my score to 6.

---

> > ### Author Response · Authors · 2024-11-22
> >
> > Thank you for your time and valuable comments that help improve the paper. Wishing you all the best!

---

### Official Review · Reviewer_BpiW · 2024-11-02

**Soundness:** 3
**Presentation:** 3
**Contribution:** 3
**Rating:** 6
**Confidence:** 4

**Summary:**

1.This paper proposes AURORACAP, a video captioner based on a large multimodal model. AURORACAP follows a simple architecture design without additional parameters for temporal modeling. To address the overhead caused by lengthy video sequences, it implements a token merging strategy, reducing the number of input visual tokens with little performance loss. AURORACAP shows superior performance on various video and image captioning benchmarks.
2.Existing video caption benchmarks only include simple descriptions, so the authors develop VDC, a video detailed captioning benchmark with over one thousand carefully annotated structured captions.
3.The authors also propose a new LLM-assisted metric VDCSCORE for better evaluation, which transforms long caption evaluation into multiple short question-answer pairs.

**Strengths:**

1.the proposed AURORACAP shows good performance. Using bipartite soft matching method to merge similar token is novel and show potential on token reduction.
2.VDC with detailed video caption is good contribution for video captioning research community, and authors provide the detailed description of dataset curation.
3.the proposed metric VDCscore is a good improvement for captioning task evaluation with novelty.
4.Paper is in good writing for understanding.

**Weaknesses:**

1.AURORACAP: there is no description of the AURORACAP's 1.3M pretraining data. We could't conclude the reseasons for the good peroformance of AUROREACAP. Could authors provide details on:

a. The sources and types of data included in the 1.3M pretraining dataset

b. Any preprocessing or filtering steps applied to this data

c. How this pretraining data compares to datasets used by other models?

This information would help readers better understand the factors contributing to AURORACAP's performance.

2.VDCSCORE: the potential stability issue of the metric.

a. Is there a fixed number of question-answer pairs generated for each caption, or does it vary?

b. If it varies, how do the authors ensure consistency in the metric across captions of different lengths or complexities?

c. Did the authors conduct any experiments to test the stability of the metric with varying numbers of question-answer pairs?


3.VDCSCORE: miss detailed information in the Elo Ranking. Could authors provide details on:

a. The specific dataset used for the Elo Ranking comparison, including its size and composition

b. The criteria used for selecting this dataset for the comparison

c. How this dataset relates to or differs from the VDC benchmark?


4.VDCSCORE：some quality problems of caption are about the repetition or grammar errors. Does this metric could evaluate these types of quality?

a.How does VDCSCORE handle linguistic aspects of caption quality, such as repetition and grammatical errors?

b.Were any specific measures incorporated into the metric to detect and penalize such issues?

c.Could the authors provide examples or experiments demonstrating how the metric performs on captions with these types of linguistic problems compared to human evaluation?

**Questions:**

the questions are listed above.

---

> ### Author Response · Authors · 2024-11-16
> **Response to Weaknesses 1- 3**
>
> Thank you for your constructive comments. We have carefully considered your suggestions and would like to provide the following
> clarifications. We have also updated the manuscript, with the modified contents highlighted in orange for your review.
>
> ---
> **Q1: AURORACAP: there is no description of the AURORACAP's 1.3M pretraining data. We could't conclude the reseasons for the good peroformance of AUROREACAP. This information would help readers better understand the factors contributing to AURORACAP's performance. Could authors provide details on: The sources and types of data included in the 1.3M pretraining dataset. Any preprocessing or filtering steps applied to this data. How this pretraining data compares to datasets used by other models?**
>
> A1: Thank you for your concern regarding the training data. Table E3 on page 32 (Table G6 on page 34 in the current version) provides details on the sources of the training data, which mostly consist of high-quality image captions. We did not perform strict filtering or cleaning of this data, as it was only used to initialize the weights of the projection layer, so the quality of the data is not particularly crucial. It is worth noting that we describe in Table D2 on page 29 (Table F5 on page 31 in the current version) how we select the pretrained ViT and LLM through pretraining data loss. Other advanced LMMs have also used similar data compositions for training.
>
> ---
> **Q2: VDCSCORE: the potential stability issue of the metric. Is there a fixed number of question-answer pairs generated for each caption, or does it vary? If it varies, how do the authors ensure consistency in the metric across captions of different lengths or complexities? Did the authors conduct any experiments to test the stability of the metric with varying numbers of question-answer pairs?**
>
> A2: Thank you for your questions regarding the stability of VDCScore. The number of question-answer pairs for each case remains approximately consistent. In Appendix M and N, we provide detailed statistics of VDCScore, highlighting that each section of the structured captions includes a similar number of question-answer pairs and outlining the distribution of question categories. We add Appendix L, Q and R for the templates used for generating these pairs. We prompt GPT-4o to generate 20 question-answer pairs per caption. To ensure consistency, we manually review and refine the generated pairs to address instances where GPT-4o fails to produce the required number or introduces hallucinations.
>
> We also conduct additional experiments by using VDCScore with varying the number of question-answer pairs. We found that as the number of question-answer pairs decreased from 20 to 5, the Pearson correlation coefficient dropped from 0.86 to 0.69. This indicates that VDCScore is less reliable with fewer question-answer pairs and that a larger number enhances its stability and reliability, thereby validating our preset number of QA pairs. We update the results in Appendix J.
>
> **Pearson correlation between VDCscore with different QA numbers and human ELO values.**
> |#QA|20|10|5|
> |-|-|-|-|
> |Pearson|0.86|0.72|0.69|
>
> ---
> **Q3: VDCSCORE: miss detailed information in the Elo Ranking. Could authors provide details on: The specific dataset used for the Elo Ranking comparison, including its size and composition. The criteria used for selecting this dataset for the comparison. How this dataset relates to or differs from the VDC benchmark?**
>
> A3: Thank you for your interest in the ELO calculation, which is an important standard for evaluating metric reliability in the paper. In Appendix O, we describe the ELO calculation method and collection process. We did not use an external dataset; instead, this process was conducted within VDC. Briefly, the steps are as follows:
>
> 1. For all models participating in the ELO ranking, we collected the output captions for each video.
>
> 2. As shown in Figure O17, we developed a frontend tool to randomly select captions generated by two different models for the same video. Without revealing the model IDs, human evaluators then chose the caption they found better.
>
> 3. We record comparison results and calculated the ELO values based on the parameters in Table O14.
>
> 4. We calculated the Pearson correlation between different metrics and human ELO values, ultimately showing that our proposed VDC metric is the closest to human evaluations.
>
> We update these descriptions in the latest manuscript in Appendix O on page 41. Thank you for your suggestions and questions.

---

> ### Author Response · Authors · 2024-11-16
> **Response to Weaknesses 4**
>
> **Q4: VDCSCORE：some quality problems of caption are about the repetition or grammar errors. Does this metric could evaluate these types of quality? How does VDCSCORE handle linguistic aspects of caption quality, such as repetition and grammatical errors? Were any specific measures incorporated into the metric to detect and penalize such issues? Could the authors provide examples or experiments demonstrating how the metric performs on captions with these types of linguistic problems compared to human evaluation?**
>
> A4:Thank you for your further questions about VDCscore. This is an aspect we hadn’t described in detail. Indeed, VDCscore lacks checks for grammar and word choice. However, we believe this is not a critical issue. The reason is that most current video understanding and captioning models are built on the foundation of LLMs, which rarely make grammar or word choice errors. In fact, we did not observe grammar issues in all the outputs we recorded. If you have any further concerns, feel free to ask, and we will continue to make progress.

---

> > ### Comment · Reviewer_BpiW · 2024-11-28
> >
> > My concerns have been mostly addressed. I retain my score 6.

---

> > > ### Author Response · Authors · 2024-11-28
> > >
> > > We greatly appreciate the time and effort you have devoted to this paper, as well as your valuable feedback. Wishing you all the best!

---

### Official Review · Reviewer_7VHc · 2024-11-02

**Soundness:** 3
**Presentation:** 3
**Contribution:** 3
**Rating:** 8
**Confidence:** 4

**Summary:**

The paper introduces AuroraCap, an LLM-based video captioning model that uses token merging to reduce visual tokens. AuroraCap is evaluated across multiple image and video captioning datasets. To assess models' ability to generate detailed captions, the paper also proposes a new benchmark, Video Detailed Captioning (VDC), along with an LLM-assisted metric, VDCscore. VDC includes over 1,000 annotated structured captions, while VDCscore transforms long captions into multiple short question-answer pairs. Various existing models are evaluated on the VDC benchmark.

**Strengths:**

* The presentation of the paper is clear. It's easy to follow.
* The proposed VDC benchmark is a valuable contribution to the community. The detailed video captioning capability of video LLMs is an area that has been relatively underexplored, and introducing VDCscore adds a meaningful component to the evaluation pipeline. The evaluation process of VDCscore aligns well with human reasoning process.
* Extensive experiments are conducted.

**Weaknesses:**

* Certain aspects of AuroraCap are not fully explored. For instance, while it performs well on the ANet dataset in video QA tasks, it performs poorly on MSVD, which is typically an easier benchmark. What might be causing this discrepancy?
* Unfair comparison:
  * Table 2 refer to the baseline models as the SoTA methods on image captioning benchmarks under zero-shot setting. However, I believe there are other LLM-based models with stronger captioning performance. For example, the BLIP family—BLIP2 [1], which achieves a 121.6 CIDEr score on NoCaps (val set), and InstructBLIP [2], which is even stronger. It would be beneficial to clarify the criteria used to select these baselines.
  * Similar concerns apply to the video captioning and video QA task comparisons.
* I notice that the VDCscore relies heavily on the use of GPT-4o:
  * As also pointed out by other work [3], different versions of GPT can affect the results of GPT assisted evaluation heavily. Although I may have missed it, I did not see which version of GPT-4o was used for the evaluation in the paper. Standardizing the evaluation method is essential for consistency in future studies.
  * Given the multiple variables that can influence GPT-assisted evaluation, and since VDCscore already uses phrased answers, what are the potential drawbacks of using an automatic metric based on, e.g. n-gram matching, to evaluate the final score? Could a variant of VDCscore be devised that assesses final triplets (i.e., <question, correct answer, predicted answer>) without relying on GPT?
* minor: Several instances of "BELU" should be corrected to "BLEU"; small grammar errors like in line 203: "are only contains" --> "only contain"



[1] Junnan Li, Dongxu Li, Silvio Savarese, Steven Hoi. BLIP-2: Bootstrapping Language-Image Pre-training with Frozen Image Encoders and Large Language Models. ICML 2023.

[2] Wenliang Dai, Junnan Li, Dongxu Li, Anthony Tiong, Junqi Zhao, Weisheng Wang, Boyang Li, Pascale Fung, Steven Hoi. InstructBLIP: Towards General-purpose Vision-Language Models with Instruction Tuning. NeurIPS 2023.

[3] Wenhao Wu. Freeva: Offline mllm as training-free video assistant. arXiv 2024.

**Questions:**

Please see Weaknesses.

---

> ### Author Response · Authors · 2024-11-16
> **Response to Weaknesses 1-4**
>
> Thank you for your constructive comments. We have carefully considered your suggestions and would like to provide the following
> clarifications. We have also updated the manuscript, with the modified contents highlighted in orange for your review.
>
> ---
> **Q1: Certain aspects of AuroraCap are not fully explored. For instance, while it performs well on the ANet dataset in video QA tasks, it performs poorly on MSVD, which is typically an easier benchmark. What might be causing this discrepancy?**
>
> A1: Thank you for you question regarding the video QA performance. Although both ANet and MSVD involve only short answers, ANet is relatively simpler, with approximately 39.6% judgment-based questions and the rest being open-ended, while MSVD is predominantly open-ended. Longwriter[4] shows that the length of model outputs is influenced by the distribution of the supervised fine-tuning training data. AuroraCap is primarily trained on videos with detailed captions, leading it to generate longer responses and perform less effectively on short question answering.
>
> ---
> **Q2: Unfair comparison: Table 2 refer to the baseline models as the SoTA methods on image captioning benchmarks under zero-shot setting. However, I believe there are other LLM-based models with stronger captioning performance. For example, the BLIP family—BLIP2 [1], which achieves a 121.6 CIDEr score on NoCaps (val set), and InstructBLIP [2], which is even stronger. It would be beneficial to clarify the criteria used to select these baselines. Similar concerns apply to the video captioning and video QA task comparisons.**
>
> A2: Thank you for pointing out additional baselines. We will update them in Table 2 with explanations. In the BLIP-2 [1] paper's section 4.2 on Image Captioning, the authors specify their training settings: “We finetune BLIP-2 models for the image captioning task, which asks the model to generate a text description for the image’s visual content. We perform finetuning on COCO and evaluate on both the COCO test set and zero-shot transfer to NoCaps.” This makes it difficult to ensure a completely fair metric comparison. In our Table 2, we mainly list the zero-shot performance of each baseline. Besides, for Flickr, AuroraCap shows 88.9 of CIDEr compared to 78.2 for InstructBLIP [2]. We clarify this point in the updated manuscript Line 162-168. Notably, the CIDEr metric is highly impacted by zero-shot versus finetuning, as shown in VALOR [5] Table 9, where the zero-shot/finetune CIDEr on the MSVD dataset is 15.4/122.6. For video captioning, we didn’t find many papers using a zero-shot setting, as some traditional models not based on LLMs are usually finetuned on the test dataset, therefore we only compare with the zero-shot models.
>
> ---
> **Q3: The VDCscore relies on GPT-4o, and different versions can significantly impact evaluation results. I didn't see which version was used in the paper, and standardizing this is crucial for consistency. Given the variables affecting GPT-assisted evaluation, what are the drawbacks of using an automatic metric like n-gram matching for the final score? Could we create a variant of VDCscore that evaluates final triplets (<question, correct answer, predicted answer>) without GPT?**
>
> A3: Thank you for pointing out this issue; we agree that it is an important one. Regarding the version of GPT-4o used, we are using the September release of the latest GPT-4o version. We will standardize and open-source all evaluation pipelines for the community’s convenience. We also greatly appreciate your suggestion to avoid using GPT for evaluating the final triplets! We missed this idea because we used LLMs to build QA pairs, so naturally continued with LLMs for evaluation. However, using an n-gram-based method is a great idea! We also test the use of n-gram metrics ROUGE to evaluate the final triplets as shown in table below and further updated in paper. We found that n-gram based metrics is less reliable than LLM evaluation as the pearson correlation 0.74 compared to VDCscore as 0.86. All these results are added in Appendix C.
>
> ---
> **Q4: Several instances of "BELU" should be corrected to "BLEU"; small grammar errors like in line 203: "are only contains" --> "only contain"**
>
> A4: Thank you for your meticulous corrections on typos and grammar. We revise these parts in the updated manuscript in Line 196-203.
>
> ---
> **Reference**
>
> [1] Junnan Li, et al. BLIP-2: Bootstrapping Language-Image Pre-training with Frozen Image Encoders and Large Language Models.
>
> [2] Wenliang Dai, et al. InstructBLIP: Towards General-purpose Vision-Language Models with Instruction Tuning.
>
> [3] Wenhao Wu. Freeva: Offline mllm as training-free video assistant.
>
> [4] Bai Y, et al. Longwriter: Unleashing 10,000+ word generation from long context llms[J].
>
> [5] Chen, Sihan, et al. "Valor: Vision-audio-language omni-perception pretraining model and dataset."

---

> ### Author Response · Authors · 2024-11-16
> **Follow-up ROUGE-based VDCScore Tables for Q3**
>
> **Pearson correlation between different metrics and human ELO values. We found that VDCScore is the closest to human ELO values, and VDCScore ROUGE falls short.**
> | Metric      | VDCScore | VDD  | ROUGE | VDCScore ROUGE |
> |-|-|-|-|-|
> | Pearson     | 0.86     | 0.12 | 0.66  |     0.74       |
>
> ---
> **Comparison of AuroraCap with LLM-based baseline methods on VDC under zero-shot structured captions setting. We consider the ROUGE based VDCScore as the evaluation metric.**
> | Model                  |  AVG.  | Camera | Short  | Background | Main Object | Detailed |
> |-|-|-|-|-|-|-|
> | **Gemini-1.5 Pro**     |  7.59  |  8.78  |  6.25  |   7.29     |    7.70     |   7.94   |
> | **LLaMA-VID**          |  6.66  |  9.53  |  5.12  |   5.30     |    6.07     |   7.29   |
> | **Video-ChatGPT-7B**   |  6.85  |  9.20  |  5.37  |   6.40     |    6.29     |   6.97   |
> | **ViLA-7B**            |  6.68  |  7.88  |  5.16  |   6.29     |    6.35     |   7.74   |
> | **Video-LLAVA-7B**     |  6.79  |  8.67  |  5.25  |   6.09     |    6.64     |   7.28   |
> | **LLAVA-1.5-7B**       |  6.34  |  8.31  |  5.40  |   5.80     |    5.52     |   6.65   |
> | **LongVA-7B**          |  6.70  |  8.37  |  4.99  |   6.11     |    6.79     |   7.23   |
> | **LLAVA-1.5-13B**      |  6.34  |  8.64  |  5.55  |   5.57     |    5.15     |   6.79   |
> | **LLAVA-NeXT-V7B**     |  6.69  |  8.61  |  5.30  |   6.35     |    6.27     |   6.93   |
> | **LLAVA-1.6-7B**       |  6.45  |  8.08  |  5.38  |   5.79     |    5.82     |   7.20   |
> | **LLAVA-1.6-13B**      |  6.18  |  7.31  |  5.28  |   6.00     |    5.29     |   7.01   |
> | **ShareGPT4Video-8B**  |  6.78  |  8.46  |  5.73  |   5.82     |    6.54     |   7.36   |
> | **LLAVA-OV-7B**        |  6.59  |  8.09  |  5.66  |   5.86     |    5.58     |   7.75   |
> | **InternVL-2-8B**      |  6.82  |  8.14  |  5.63  |   5.95     |    6.53     |   7.86   |
> | **AURORACAP-7B**       |  **7.60**  |  9.55  |  5.66  |   6.79     |    7.42     |   8.58   |
>
> ---
> **Comparison of AuroraCap with LLM-based baseline methods on both LLM-based VDCScore and ROUGE based VDCScore, and other evaluation metrics under zero-shot setting. For each evaluation metric, we report the average value of the five structured captions in VDC. Note that VDD, CIDEr, and BELU are only the average of background and main object caption, since the values of the others are closed to zero.**
> | Model | VDCScore Acc | VDCScore Score | VDCScore ROUGE | VDD Acc | VDD Score | C    | B@1  | B@4  | M    | R    | Elo  |
> |-|-|-|-|-|-|-|-|-|-|-|-|
> | **Gemini-1.5 Pro**       | 41.73        | 2.15           |    7.59        | 49.68   | 3.07    | 5.97 | 29.72 | 2.63 | 21.21 | 20.19 | 1,171 |
> | **LLaMA-VID**            | 30.86        | 1.62           |    6.66        | 4.63    | 1.63    | 1.48 | 17.74 | 1.46 | 8.07  | 17.47 | 859   |
> | **Video-ChatGPT-7B**     | 31.12        | 1.62           |    6.85        | 8.57    | 1.84    | 2.92 | 17.31 | 2.19 | 11.57 | 16.96 | 944   |
> | **MovieChat-7B**         | 31.92        | 1.64           |    6.68        | 10.24   | 1.86    | 5.14 | 14.33 | 3.17 | 13.60 | 14.98 | 890   |
> | **VILA-7B**              | 32.61        | 1.70           |    6.79        | 16.27   | 2.02    | 8.20 | 19.13 | 2.11 | 5.62  | 16.63 | 1,073 |
> | **Video-LLAVA-7B**       | 32.80        | 1.72           |    6.34        | 14.14   | 2.00    | 4.43 | 17.20 | 2.32 | 10.36 | 17.53 | 1,007 |
> | **LLAVA-1.5-7B**         | 33.98        | 1.76           |    6.34        | 26.71   | 2.33    | 6.63 | 29.80 | 2.54 | 22.79 | 20.36 | 825   |
> | **LongVA-7B**            | 34.50        | 1.79           |    6.70        | 32.65   | 2.69    | 4.83 | 18.75 | 2.16 | 13.43 | 14.84 | 969   |
> | **LLAVA-1.5-13B**        | 34.78        | 1.80           |    6.34        | 28.26   | 2.36    | 3.90 | 20.43 | 2.02 | 26.37 | 17.87 | 943   |
> | **LLAVA-NeXT-V7B**       | 35.46        | 1.85           |    6.69        | 25.62   | 2.34    | 2.66 | 20.18 | 2.33 | 28.17 | 17.51 | 1,022 |
> | **LLAVA-1.6-7B**         | 35.70        | 1.85           |    6.45        | 40.16   | 2.69    | 3.09 | 17.36 | 1.59 | 24.23 | 17.08 | 846   |
> | **LLAVA-1.6-13B**        | 35.85        | 1.85           |    6.18        | 34.55   | 2.51    | 5.55 | 29.23 | 2.50 | 20.26 | 19.96 | 728   |
> | **ShareGPT4Video-8B**    | 36.17        | 1.85           |    6.78        | 36.44   | 1.85    | 1.02 | 12.61 | 0.79 | 8.33  | 16.31 | 1,102 |
> | **LLAVA-OV-7B**          | 37.45        | 1.94           |    6.59        | 41.83   | 2.70    | 4.09 | 28.34 | 2.84 | 23.98 | 19.59 | 1,155 |
> | **InternVL-2-8B**        | 37.72        | 1.96           |    6.82        | 48.99   | 3.03    | 5.59 | 15.75 | 2.48 | 10.76 | 17.63 | 1,081 |
> | **AURORACAP-7B (ours)**  | 38.21        | 1.98           |    7.60        | 48.33   | 2.90    | 9.51 | 30.90 | 4.06 | 19.09 | 21.58 | 1,267 |

---

> > ### Comment · Reviewer_7VHc · 2024-11-22
> >
> > Thank you for the detailed response. The new results are impressive and further enhance the paper's contributions. My concerns have been fully addressed, and I am updating my score to 8.

---

> > > ### Author Response · Authors · 2024-11-22
> > >
> > > Thanks for your time and careful review. Wishing you all the best!

---

### Official Review · Reviewer_K52W · 2024-11-04

**Soundness:** 3
**Presentation:** 3
**Contribution:** 4
**Rating:** 8
**Confidence:** 4

**Summary:**

Towards the video detailed captioning task, this paper conducts comprehensive research including the introduction of a new model called AuroraCap, a novel dataset VDC and a novel evaluation metric VDC-Score. For AuroraCap, authors apply the token merging strategy on large models. Extensive experiments have been conducted to verify the effectiveness of AuroraCap, and the necessity of VDC and VDCScore. This work contributes significantly to advancing the field of detailed video understanding.

**Strengths:**

1.	This paper is well-written.
2.	The proposed benchmark VDC is large-scale and diverse, which contains short, background, main object, camera and detailed captions.
3.	The average length of captions in VDC exceeds previous datasets, to facilitate the research of understanding and also generation of long videos.
4.	Very extensive experiments.

**Weaknesses:**

1. The paper is not very easy to follow, owing to many tables and figures.
2. For all the compared captioning models, are they trained under the same schema? If true, which one setting is chosen among different settings presented in Figure D5 and Figure D6?
3. In Table 6 – Table 8, what does the accuracy mean?
4. For VDCScore, it is hard to evaluate which score is better among VDCScore, VDD, CIDEr and maybe CLIP-based score, RefCLIP-S[1] and RefPAC-S[2].

[1] Hessel, Jack, et al. "Clipscore: A reference-free evaluation metric for image captioning." arXiv preprint arXiv:2104.08718 (2021).

[2] Sarto, Sara, et al. "Positive-augmented contrastive learning for image and video captioning evaluation." Proceedings of the IEEE/CVF conference on computer vision and pattern recognition. 2023.

**Questions:**

1. How many GPU hours have been consumed on this research?
2. Will you publish the data, the pre-trained model with weights and also the training recipes?
3. In the future, I believe the VDC benchmark will facilitate more research on the video, including classification, video generation, etc.

---

> ### Author Response · Authors · 2024-11-16
> **Response to Weaknesses 1-4**
>
> Thank you for your constructive comments. We have carefully considered your suggestions and would like to provide the following
> clarifications. We have also updated the manuscript, with the modified contents highlighted in orange for your review.
>
> ---
> **Q1: The paper is not very easy to follow, owing to many tables and figures.**
>
> A1: Thank you for evaluating the structure of this paper from a reader's perspective. We enhance the reading experience by providing a more detailed explanation of the article's structure in the introduction of manuscript Line 111-116. Broadly, the paper is divided into two parts. Section 2 covers the model architecture and training process, along with comparative results of the model's performance on various existing benchmarks. In Section 3, we discuss how a new video detailed captioning benchmark are constructed and compare the performance of accessible models on this new benchmark. At the end of the main text, we analyze the rationale behind some of our metrics and conduct an ablation study on the VDC benchmark. Due to space constraints, we have included additional ablation studies of models on existing datasets in Appendix F to provide further insights. The above summarizes the data related to our figures and tables.
>
> ---
> **Q2: For all the compared captioning models, are they trained under the same schema? If true, which one setting is chosen among different settings presented in Figure D5 and Figure D6?**
>
> A2: Thank you for your question regarding training details. In fact, the baseline models mentioned in the paper (such as LLava, etc.) have their own distinct architectures and training data. Since these are methods proposed in other papers, it is not possible for us to conduct a fair comparison. However, in Figures D5 and D6, we primarily perform an ablation study on our proposed AuroraCap, ensuring a fair comparison. For Figure D5, we focus on the efficiency of different inference strategies; the detailed settings can be found in the caption of Figure D5. For Figure D6, we are concerned with the efficiency and performance of different training strategies, and on middle of page 28, we describe the settings for training methods A-E. Overall, the content of these two figures is intended to demonstrate that the token merging strategy improves efficiency in both training and inference of video LMMs while maintaining performance without significant degradation.
>
> ---
> **Q3: In Table 6 – Table 8, what does the accuracy mean?**
>
> A3: Thank you for your question regarding the accuracy. For each case, we first divide the ground truth caption into several short question-answer pairs, and obtain predictions from the generated captions with the assistance of GPT-4o (We add the prompt templare in new Appendix section). We utilize GPT-4o to evaluate the correctness and assign a score for each question-answer pair, reporting correctness as ‘yes’ or ‘no’ and providing a score on a scale from 0 to 5. We calculate both the accuracy and the average score for each individual case, and then compute the overall average accuracy and average score across all cases in the VDC to present the final results. The prompt template used to assist GPT-4o is identical to that proposed by VideoChatGPT[3]. We add the prompt templare in new Appendix section. We update the manuscript on page 43-44 with these clarifications to ensure a better understanding of the accuracy and our evaluation method.
>
> ---
> **Q4: For VDCScore, it is hard to evaluate which score is better among VDCScore, VDD, CIDEr and maybe CLIP-based score, RefCLIP-S[1] and RefPAC-S[2].**
>
> A4: Thank you for your insightful suggestion. To ensure the fairness of VDCScore, we conduct human Elo ranking as shown in the last column of Table 6, and calculated the Pearson correlation coefficients between Elo rankings and various metrics. As depicted in Figure 5, VDCscore shows the better correlation with human evaluation results. During our experiments, we found that CIDEr consistently yields near-zero results for detailed caption evaluation, making it ineffective for distinguishing caption quality. VDCScore is designed to measure the alignment between ground truth and predicted detailed captions without requiring visual input，while RefCLIP-S[1] and RefPAC-S[2] focus on short caption evaluations with visual input. Their performance may diminish with longer texts because of the short image-text training data of CLIP. Additionally, we have calculated the CLIP similarity between ground truth and predicted captions and its Pearson correlation with Elo rankings. We found that the correlation between CLIP score and human ratings is low, which may be due to the fact that, in a lengthy caption, there may be only one or two fine-grained errors that the CLIP text encoder lacks the capability to detect. We add the discussion in appendix section J on page 38.
>
> |Metric|VDCScore|VDD|ROUGE|CLIP-based|
> |-|-|-|-|-|
> |Pearson Correlation|0.86|0.12|0.66|0.39|

---

> ### Author Response · Authors · 2024-11-16
> **Response to Questions 1 - 3**
>
> **Q5: How many GPU hours have been consumed on this research?**
>
> A5: Thank you for your concern regarding training costs. In the last column of Figure F8, we have listed the total GPU hours required for each training setup. Frankly, this is not a small number, and due to parameter tuning, data testing, and comparative experiments, the overall estimated training time exceeds 10,000 H100 hours for this project. Fortunately, based on our final trials, we believe that under training setting C for the best performance-efficiency trade-off, a single complete training run requires approximately 700 H100 hours.
>
> ---
> **Q6: Will you publish the data, the pre-trained model with weights and also the training recipes?**
>
> A6: Thank you for your interest in the open-source availability. Yes! We are committed to releasing the training and deployment code, model weights, training data, and a one-line test script for the benchmark. We also pledge to release additional video data that has been re-captioned by our model. Due to the double-blind review setting, we are currently unable to share further details. We believe that open-sourcing will benefit the community and bring broader attention to this task.
>
> ---
> **Q7: In the future, I believe the VDC benchmark will facilitate more research on the video, including classification, video generation, etc**
>
> A7: Thank you for your kind words. We also believe that video detailed captioning is fundamental to video understanding and generation tasks.
>
> ---
> **Reference**
>
> [1] Hessel, Jack, et al. "Clipscore: A reference-free evaluation metric for image captioning." arXiv preprint arXiv:2104.08718 (2021).
>
> [2] Sarto, Sara, et al. "Positive-augmented contrastive learning for image and video captioning evaluation." Proceedings of the IEEE/CVF conference on computer vision and pattern recognition. 2023.
>
> [3] Maaz M, Rasheed H, Khan S, et al. Video-chatgpt: Towards detailed video understanding via large vision and language models[J]. arXiv preprint arXiv:2306.05424, 2023.

---

> > ### Comment · Reviewer_K52W · 2024-11-22
> >
> > My concerns have been addressed. I retain my score.

---

> > > ### Author Response · Authors · 2024-11-22
> > >
> > > Thank you for recognizing our work. Wishing you all the best!

---

### Meta-Review · Area_Chair_4Mhh · 2024-12-16

**Metareview:**

This paper introduces a new benchmark and evaluation metric for video detailed captioning, alongside a strong baseline model AuroraCap with token merging strategy. Reviewers acknowledged the clear presentation and contributions of the proposed benchmark, metric, and extensive experiments. The initial concerns involved the unclear explanation of the token merging, missing training details, unfair comparisons and questions on the metric. The rebuttal well addressed these concerns, leading all five reviewers to unanimously recommend acceptance. The AC thus suggests accepting the paper and the authors should follow the reviewers’ comments to revise the final version.

**Additional Comments On Reviewer Discussion:**

The initial concerns involved the unclear explanation of the token merging, missing training details, unfair comparisons and questions on the metric. The rebuttal well addressed these concerns, leading all five reviewers to unanimously recommend acceptance.

---

### Decision · Program_Chairs · 2025-01-22

Accept (Poster)